# Detection of Circulating Tumor DNA in Liquid Biopsy: Current Techniques and Potential Applications in Melanoma

**DOI:** 10.3390/ijms26020861

**Published:** 2025-01-20

**Authors:** Clara Martínez-Vila, Cristina Teixido, Francisco Aya, Roberto Martín, Europa Azucena González-Navarro, Llucia Alos, Natalia Castrejon, Ana Arance

**Affiliations:** 1Department of Medical Oncology, Althaia Xarxa Assistencial Universitària de Manresa, Dr. Joan Soler, 1–3, 08243 Manresa, Spain; cmartinezv@althaia.cat; 2Programa de Doctorat en Medicina i Recerca Translacional, Facultat de Medicina, Universitat de Barcelona, 08036 Barcelona, Spain; 3Institut de Recerca i Innovació en Ciències de la Vida i de la Salut a la Catalunya Central (IRIS-CC), Roda 70, 08500 Vic, Spain; 4Department of Pathology, Hospital Clínic of Barcelona, University of Barcelona, Villarroel 170, 08036 Barcelona, Spain; teixido@clinic.cat (C.T.); lalos@clinic.cat (L.A.); castrejon@clinic.cat (N.C.); 5August Pi i Sunyer Biomedical Research Institute (IDIBAPS), Rosselló 149, 08036 Barcelona, Spain; faya@clinic.cat (F.A.); robertomartinhuertas@outlook.com (R.M.); aegonzal@clinic.cat (E.A.G.-N.); 6Department of Medical Oncology, Hospital Clínic of Barcelona, University of Barcelona, Villarroel 170, 08036 Barcelona, Spain; 7Department of Immunology, Hospital Clínic of Barcelona, University of Barcelona, Villarroel 170, 08036 Barcelona, Spain

**Keywords:** melanoma, biomarker, ctDNA, disease monitoring, mutational landscape, ICI

## Abstract

The treatment landscape for advanced melanoma has transformed significantly with the advent of BRAF and MEK inhibitors (BRAF/MEKi) targeting *BRAF*V600 mutations, as well as immune checkpoint inhibitors (ICI) like anti-PD-1 monotherapy or its combinations with anti-CTLA-4 or anti-LAG-3. Despite that, many patients still do not benefit from these treatments at all or develop resistance mechanisms. Therefore, prognostic and predictive biomarkers are needed to identify patients who should switch or escalate their treatment strategies or initiate an intensive follow-up. In melanoma, liquid biopsy has shown promising results, with a potential role in predicting relapse in resected high-risk patients or in disease monitoring during the treatment of advanced disease. Several components in peripheral blood have been analyzed, such as circulating tumor cells (CTCs), cell-free DNA (cfDNA), and circulant tumoral DNA (ctDNA), which have turned out to be particularly promising. To analyze ctDNA in blood, different techniques have proven to be useful, including digital droplet polymerase chain reaction (ddPCR) to detect specific mutations and, more recently, next-generation sequencing (NGS) techniques, which allow analyzing a broader repertoire of the mutation landscape of each patient. In this review, our goal is to update the current understanding of liquid biopsy, focusing on the use of ctDNA as a biological material in the daily clinical management of melanoma patients, in particular those with advanced disease treated with ICI.

## 1. Introduction

Melanoma results from the accumulation of several genetic alterations induced by UV radiation-induced cellular damage. The genes most frequently associated with melanoma pathogenesis include *BRAF*, *NRAS*, *neurofibromin 1* (*NF1*), and *KIT* [1]. Activating *BRAF* mutations, primarily at codon 600 with V600E as the most prevalent change, are found in about 40–60% of melanomas. In contrast, oncogenic *NRAS* mutations, which do not co-occur with *BRAF* mutations, are present in approximately 15–25% of melanoma cases [2,3]. Particularly in the setting of advanced melanoma (MM), significant improvement has been made in treatment strategies over the last decade. Initially, ipilimumab, a human IgG1 monoclonal antibody directed against CTLA-4, demonstrated superiority over chemotherapy (ChT) in terms of progression-free survival (PFS) and overall survival (OS) [4,5]. However, later on, anti-PD-1 monoclonal antibodies, pembrolizumab and nivolumab, were approved for the treatment of MM, based on pivotal trials that showed anti-PD-1 monotherapy achieved an objective response rate (ORR) of 35 to 42% and a 7-year OS of 37.8% [6,7], overcoming results from anti-CTLA-4 monotherapy.

Subsequently, the combination of anti-PD-1 and anti-CTLA-4 has shown even better results, with an ORR of around 50–70% and a 5-year OS of 59–68% [8]. Recently reported data from the pivotal phase III trial with a minimum follow up of 10 years reported a median OS of 71.9 months with nivolumab plus ipilimumab, 36.9 months with nivolumab, and 19.9 months with ipilimumab, with a 10-year OS of 43% [9] (Table 1). 

Over time, other molecules that inhibit other immune checkpoints have emerged and have also shown superiority to anti-PD-1 monotherapy. This is the case with the combination of anti-PD-1 (nivolumab) with anti-LAG-3 (relatlimab) in the first-line treatment of advanced melanoma. In a recent first-line phase III pivotal trial with a median follow-up of 19 months, the results demonstrated a median PFS of 10.2 months for combination therapy compared to 4.6 months with nivolumab alone. The median OS had not yet been reached (NR) for the combination, whereas it took 34.1 months with nivolumab alone. Additionally, the ORR was 43.1% for the combination therapy, surpassing the 32.6% achieved with nivolumab monotherapy [10].

To date, the two anti-PD-1 combination strategies have not been directly compared, but the combination of anti-PD-1 with anti-LAG-3 appears to have a lower rate of grade 3 or higher immune-related adverse events (irAE) [8,9,10].

Finally, although not the main focus of our review, it is also important to mention targeted therapy with BRAF inhibitors (BRAFi) in combination with MEK inhibitors (MEKi) as treatment option in *BRAF*V600-mutant MM patients, either as first-line or second-line therapy [11]. Currently, three different BRAF/MEKi combinations have demonstrated an ORR of 64–69% and a 5-year OS rate of approximately 34% in their pivotal trials, all of which have been approved by the FDA and EMA in the setting of advanced melanoma. Although these combinations have not been directly compared in randomized trials, they have slightly different toxicity profiles [12,13,14] (Table 1).

These therapies that have proven to be effective in advanced melanoma have also shown benefit in the (neo)adjuvant setting for resectable stages (IIB-IV). Anti-PD-1 therapies have become the standard of care for resectable stage IIB-IV melanoma based on the results of pivotal clinical trials (Table 1).

The KEYNOTE-716 trial evaluated pembrolizumab in patients with stage IIB-IIC melanoma. At 36 months, recurrence-free survival (RFS) was 76.2% with pembrolizumab in comparison to 63.4% with placebo [15]. Similarly, the CheckMate 76 K trial showed that adjuvant nivolumab significantly improved RFS compared to placebo [16]. Although OS data are not yet available, pembrolizumab and nivolumab are approved for adjuvant use in Europe (Table 1).

In the CheckMate 238 trial, nivolumab showed a significant RFS benefit over ipilimumab in stage IIIB-IV American Joint Committee on Cancer 7^th^ edition (AJCCv7) melanoma [17,18], while the KEYNOTE-054 trial showed both RFS and distant metastasis-free survival (DMFS) benefits at 5 years with pembrolizumab in stage III patients in comparison to placebo [19].

In patients with stage III *BRAF*V600-mutant melanoma, adjuvant BRAF/MEKi therapy has shown activity. In the COMBI-AD trial, adjuvant dabrafenib-trametinib for 12 months in comparison to placebo improved RFS and DMFS in resected AJCCv7 stage III (sentinel lymph node (SLN) > 1 mm) melanoma with *BRAF V600E/K* mutations, although at the final analysis (>10 years of follow-up), the median OS was NR in either arm (*p* = 0.06), and in the subgroup analysis, the OS benefit was limited in patients with the V600K mutation [20]. Based on these results, EMA approved dabrafenib-trametinib adjuvant treatment for resectable stage III *BRAF*V600E-mutant melanoma, but it has not been authorized in Spain (Table 1).

Promising results have also been seen in the neoadjuvant setting, with anti-PD-1 monotherapy and combinations of anti-PD-1 and anti-CTLA-4. The SWOG S1801 trial showed that neoadjuvant pembrolizumab combined with adjuvant therapy prolonged event-free survival (EFS) compared with adjuvant therapy alone [21]. In addition, the NADINA trial showed that neoadjuvant nivolumab plus ipilimumab for two cycles followed by adjuvant treatment (nivolumab or dabrafenib and trametinib in *BRAF*V600mutant melanoma) improved EFS and DMFS compared to adjuvant nivolumab in resectable stage III melanoma [22] (Table 1).

Although these (neo)adjuvant strategies are not yet approved by the EMA or FDA, they have been included in updated ESMO guidelines, reflecting their growing role in the comprehensive management of resectable melanoma [23] (Table 1).

**Table 1 ijms-26-00861-t001:** Clinical characteristics, prognosis and current management in different melanoma stages (based on [21,23]).

Stage and TNM AJCCv8	Melanoma Specific Survival (MSS) 5 Years and 10 Years	Standard Local Treatment	(Neo)Adjuvant or Advanced Setting Treatment
I-IIA (pTbN0-pT3a)	IA 5 y 99% 10 y 98% IB 5 y 97% 10 y 94% IIA 5 y 94% 10 y 88%	WLE of primary plus SLN dissection. CLND is not recommended for patients with a positive SLN. Standard follow up	Clinical trial
IIB-IIC (T3b-T4bN0)	IIB 5 y 87% 10 y 82% IIC 5 y 82% 10 y 75%	WLE of primary plus SLN dissection. CLND is not recommended for patients with a positive SLN.	Adjuvant therapy with either pembrolizumab or nivolumab for 12 months should be considered. Clinical trial
Resectable IIIA-IIID -IV	IIIA 5 y 93% 10 y 88% IIIB 5 y 83% 10 y 77% IIIC 5 y 69% 10 y 60% IIID 5 y 32% 10 y 24%	WLE of primary CLND is not recommended for patients with a positive SLN. Patients with resectable ITMs should undergo WLE Stage III: upfront resection or after neoadjuvant treatment Resectable stage IV: Metastasectomy or local ablative	Adjuvant anti-PD-1 therapy (nivolumab for resected stage IIIB-IV or pembrolizumab for resected stage III) or dabrafenib and trametinib for patients with resected stage III *BRAF*V600E-mutant melanoma (not authorized in Spain). For patients with AJCC8 stage IIIA and SLN < 1 mm, adjuvant treatment is generally not recommended. Other options not EMA or FDA approved: Neoadjuvant nivolumab plus ipilimumab followed by adjuvant therapy based on pathological response and *BRAF* status. Neoadjuvant plus adjuvant pembrolizumab. Clinical trial
Non-resectable III and IV	IV OS 5 y 59–68% 10 y 43%		First-line Ipilimumab and nivolumab is a preferred option for all patients regardless of *BRAF* status. First-line nivolumab or pembrolizumab is also recommended. BRAF/MEKi combination therapy is also an option in the first line for patients with *BRAF*V600-mutant melanoma. Clinical trial

Abbreviations: AJCCv8, AJCC 8th edition; CLND, complete lymph node dissection; ITMs, in-transit metastases; WLE, whole local excision; y, years.

However, despite all this progress, not all patients respond to these treatment strategies, and the emergence of resistance in many patients is a huge challenge. Therefore, robust biomarkers to identify treatment benefit or the emergence of resistance are critical. Currently, the most investigated biomarker as a predictor for anti-PD-1 response is tumor PD-L1 expression, determined by immunohistochemistry (IHC) in tumor tissue in lung, urothelial, and gastric cancers, but its low specificity, 62–72% across trials [24,25,26], is a limitation since approximately 20% of melanoma patients with negative expression also benefit from anti-PD-1 treatment [7,9]. Other potentially useful biomarkers for predicting response to anti-PD-1 have been described, including an increase in CD8+ T cell density from baseline to early treatment initiation and baseline intratumoral PD-1+ T cell density, which both seem to correlate with anti-PD-1 response [27].

In addition, elevated serum lactate dehydrogenase (LDH), which is only considered a non-predictive prognostic factor in melanoma, is an enzyme involved in anaerobic metabolism and is therefore non-specific and can be elevated in a variety of benign and malignant conditions. However, it has been shown to discriminate quite accurately poor-prognosis melanoma patients treated with ICI and BRAF/MEKi in clinical trials [28].

Finally, liquid biopsy, in particular ctDNA, represents a promising non-invasive method in the management of melanoma [29] and could be a valuable tool as a prognostic and predictive biomarker, firstly, in high-risk resected patients (AJCCv8 stage IIB/C, III and IV [30]) to detect minimal residual disease (MRD) and predict their potential risk of relapse and, secondly, but no less importantly, to monitor initial and long-term response, emerging resistance and progression in patients with advanced melanoma disease (Figure 1).

The term liquid biopsy encompasses various biological components present in blood or other biological fluids such as pleural fluid, ascitic fluid or cerebrospinal fluid (CSF). We distinguish the following: CTCs, cfDNA and cfRNA, platelet-associated RNA, exosomes and non-coding microRNA (miRNA) [31]. cfDNA can be readily extracted from serum and plasma, containing DNA fragments derived from tumors, known as ctDNA [32] (Figure 2).

The main focus of this review is to detail the potential applications of ctDNA as a biological material in the daily clinical practice of patients with advanced melanoma treated with ICI (Figure 1 and Figure 2).

### 1.1. Genomic Alterations Defining Melanoma

Approximately 85% of melanomas harbor primary pathogenic alterations in key genes such as *BRAF*, *NRAS*, *NF1* or *KIT*, which include mutations, deletions or amplifications [33].

Located in the 7q34 region of chromosome 7, the *BRAF* gene encodes a serine/threonine kinase that plays a key role in activating the MEK-ERK pathway downstream of RAS signaling [34]. The most common mutation, V600E, is found in more than 90% of melanomas and is a consequence of the T1799A transversion. Other variants, such as V600K and V600D, are less common. The V600E mutation confers constitutive kinase activity and activates the RAS-RAF-MEK pathway, leading to cell proliferation, resistance to apoptosis and tumor progression. The prevalence of the *BRAF* gene mutation is highest in advanced melanoma, where it is found in 50–60% of patients and in 30% of localized melanomas [34].

Taken together, the RAS-RAF-MEK-ERK signaling pathway and its targets are part of what is known globally as the MAPK pathway.

*NRAS*, the first proto-oncogene discovered in melanoma [3,35], encodes a small GTPase and is mutated in approximately 15–25% of melanomas [3,36], the most common mutation being Q61, mainly QR, QL and QK. In turn, mutations in other genes can lead to RAS overstimulation, such as loss-of-function mutations in *NF1*, which promote RAF protein phosphorylation and activation of its targets [2,3].

*BRAF*, *NRAS*, or *NF1* mutations can co-occur with C/T mutations in the promoter region of the telomerase reverse transcriptase (*TERT*) gene, which are frequently observed in melanoma, occurring in 30–85% of cases depending on disease stage. *TERT* encodes the catalytic subunit of telomerase, a ribonucleoprotein that maintains telomere length and is critical for cell immortality [37,38]. These mutations are the result of a transition from cytidine to thymidine in the *TERT* gene promoter on chromosome 5, called C228T and C250T, and their presence in cutaneous melanoma is associated with poor prognosis.

Finally, *KIT* encodes a receptor tyrosine kinase that is directly responsible for binding to growth factors that initiate the MAPK and PI3K-AKT pathways. Mucosal and acral melanoma are the melanoma types with the highest prevalence of *KIT* alterations, around 20–30% and 10–15%, respectively. However, only 1% of cutaneous melanomas [3,39] carry *KIT* mutations.

These genetic alterations, which are considered to be pathogenic in melanoma, can be detected in patients’ blood and allow ctDNA to be more accurately distinguished from normal circulating DNA within cfDNA (Figure 1 and Figure 2).

Furthermore, considering that the most common mutations of *BRAF*, *NRAS*, *KIT* and *NF1* are encoded by single base pair substitutions or in tandem, and they can be easily detected using primer/probe-based strategies through PCR, which will be discussed below.

### 1.2. ctDNA, Rational for Clinical Implementation in Melanoma

Historically, information about the molecular characterization of melanoma has been obtained from tumor tissue, ideally from metastatic lesions, but if this is not possible, from the primary tumor. However, solid tissue biopsies can be risky, painful, expensive, time-consuming and require a lot of time and trained medical staff. In addition, a single tissue biopsy may not represent the complexity of tumor heterogeneity, and tumor genotypes may change under the selective pressure of treatment. Repeatedly performing biopsies of tumor tissue is a crude procedure, and ctDNA offers the possibility of serial sampling to monitor disease progression and gain a broader view of tumor heterogeneity, if present [32].

Since the *BRAF* V600E mutation is the most common and significant pathogenic alteration in melanoma, efforts in this cancer have primarily been directed at detecting the V600E mutation in ctDNA. However, additional genomic alterations and a variety of detection techniques have since been explored.

### 1.3. cfDNA and ctDNA as Biological Material

ctDNA in blood was first discovered in 1948 [40], and to date, the most accepted hypothesis regarding the release of ctDNA to the bloodstream is that tumor cells that undergo apoptosis or necrosis, if not phagocytized, enter the bloodstream as ctDNA [41]. The estimated size of ctDNA varies from ~120 to 180 bp, with a peak at about 165 bp, which is characteristic of the apoptotic process [41].

In patients with advanced solid tumors, ctDNA accounts for approximately 1% of total DNA in the blood and has a short half-life, ranging from 16 min to 13 h [42].

Furthermore, the amount of ctDNA shed into the blood depends on the type of tumor, a concept known as tumor-shedding, which influences the ctDNA detection rate; in advanced *BRAF*-mutant melanoma, it is reported to be around 80%. This is also influenced by the rate at which ctDNA is released into the bloodstream, which depends on the stage, location, volume and angiogenesis component of the tumor [42]. Therefore, cfDNA concentrations can vary widely, ranging from 7 nanograms per milliliter (ng/mL) in healthy donors to as high as 1125 ng/mL in patients with advanced melanoma [42].

Due to the low concentration and short half-life of ctDNA in peripheral blood, its detection requires highly sensitive techniques.

### 1.4. Analytical Techniques for ctDNA

There are several methods for analyzing ctDNA. The methods traditionally used in tumor tissue samples, such as Sanger sequencing, pyrosequencing or quantitative PCR (qPCR), referred as standard PCR-based techniques, are not the most sensitive options available today for analyzing ctDNA [43]. Although they can identify mutated alleles, their effectiveness is limited by the overwhelming presence of non-mutated alleles in the blood. Over the last decade, more advanced mutation-specific techniques have been developed, such as BEAMing and digital PCR [44,45]. These methods have analytical sensitivities ranging from to 0.01% to 0.005%. The diagnostic sensitivity of these methods varies from 34% to 100%, and also depends on tumor type and stage of disease (Figure 2 and Table 2).

BEAMing uses magnetic beads coated with specific primers to capture specific ctDNA fragments prior to amplification [46] (Figure 3). This technique has been shown to have greater analytical sensitivity than qPCR. In studies performed on colorectal cancer patients, LoD ranging from 0.01% to 0.18% [44] and 0.01% for *BRAF* V600 in melanoma patients [47] were obtained.

Digital PCR is another approach to maximize the detection of ctDNA when it is present at low concentrations. There are several different systems of digital PCR: those that use pre-formed microchambers on a chip or those that use dynamically formed droplet partitions [48,49]. For the purpose of this review, we will only focus on the latter approach, known as ddPCR. In the ddPCR workflow, the 20 μL reaction mixture is divided into approximately 20,000 equally sized droplets in a water–oil emulsion [50]. Each droplet contains the target molecule(s), the non-target molecule(s), both or neither. The reaction is amplified to completion in a thermocycler and the fluorescence of each droplet is measured in a specialized droplet reader, similar to a flow cytometer. By dividing the sample into partitions (i.e., droplets) and analyzing them individually, the probability of detecting rare target molecules is increased (Figure 3).

ddPCR can accurately detect SNVs with LoDs of 0.005% for *BRAF* and 0.05% for *TERT* promoter mutations [51,52].

More recently, NGS approaches, such as WGS [53], WES [54], and targeted sequencing panels [55], have been evaluated also for ctDNA analysis, which covers thousands of regions in ctDNA in one assay read. Disadvantages include the costs, the time needed to process the samples and the substantial volume of resulting data to process. Furthermore, since ctDNA usually represents less than 1% of plasma in most patients with advanced disease, this strategy represents a significant challenge for any library preparation methodology required for sequencing. Its use requires specific strategies such as barcoding and target capture to reach an analytical sensitivity < 1% [56] with standard NGS (Figure 2 and Figure 3 and Table 2).

### 1.5. Emerging NGS Technologies

More recently, low standard NGS sensitivity can be improved with modified NGS, which can reach an LoD until 0.1 or even to 0.01% with specific techniques, such as amplicon deep sequencing and hybrid-capture deep sequencing. In this regard, a growing number of pan-cancer ctDNA-optimized NGS gene panels have emerged as promising technologies, including Guardant360^®^ CDx (USA) and FoundationOne^®^Liquid CDx (USA) [56,57] (Figure 2 and Figure 3 and Table 2).

Guardant360^®^ CDx is a qualitative NGS-based in vitro diagnostic device that uses targeted high-throughput hybridization-based capture technology for the detection of SNVs, insertions and deletions (indels) in 55 genes, CNA in 2 genes, and fusions in 4 genes. Guardant360^®^ CDx utilizes cfDNA from plasma of peripheral whole blood collected in two 10 mL streck cfDNA blood collection tubes [58].

FoundationOne^®^ Liquid CDx is an NGS-based in vitro diagnostic method targeting 324 genes that is approved by the FDA. It uses circulating cfDNA isolated from plasma of peripheral whole blood collected in two 10 mL anti-coagulated blood collection tubes (20 ng of extracted DNA) [59]. All coding exons of 309 genes are targeted; select intronic or non-coding regions are targeted in 21 of these genes. Additionally, select intronic or non-coding regions are targeted in 15 genes, resulting in 324 total targeted genes. The assay detects substitutions, indels, genomic rearrangements, CNAs including amplifications and losses.

Both panels are FDA-approved and have the reported ability to detect mutant allele frequencies (MAFs) as low as 0.01%.

All recommend around 20–30 ng of starting cfDNA for high-quality library preparation, and this amount of cfDNA can be commonly obtained from 20 mL of blood (10 mL of blood ≈ 4–5 mL of plasma). Less cfDNA can be used as the input but will result in a subsequent decrease in the limit of detection.

However, particularly in early-stage cancer, another challenge is the low fraction of ctDNA in cfDNA and, consequently, the presence of tumor mutations in plasma at variant allele fractions (VAFs) potentially below the background sequencing error threshold.

Moreover, these optimized NGS can detect low-frequency variants, known as variants of unknown significance (VUS), but the clinical relevance of these variants, which may reflect CHIP non-tumor cfDNA, remains unclear [56,57].

Another approach for addressing the limited abundance of cfDNA is the use of WGS to increase the breadth of sequencing. Bespoke tumor-informed ctDNA panels incorporating this strategy of probing multiple known mutations via personalized ctDNA assays have been developed [60,61]. This method consists of sequencing tumor tissue samples and selecting relevant mutations for a custom-built ctDNA assay.

Particularly relevant are those bespoke assays that compare WES or WGS from germline DNA and tumoral tissue DNA, which help to overcome the influence of CHIP in non-tumor cfDNA.

For example, a clinically focused company, Natera, offers an individualized NGS panel, known as Signatera^TM^, which is custom-built ctDNA test for already diagnosed cancer patients. It involves the selection of 16 somatic variants identified through WES of paired primary tumor and germline DNA samples, followed by the design of patient- specific assays using multiplex- PCR amplification and subsequently NGS [62]. A LoD of 0.004% have been described with Signatera^TM^ for ctDNA analysis in melanoma patients and other solid tumors treated with anti-PD-1 based therapies.

RaDaR^TM^ is a personalized ctDNA assay [63] that employs multiplex-PCR amplification combined with targeted NGS. Initially, WES is performed to identify somatic variants in tumor tissue, which are then used to design a patient-specific primer panel. This panel includes up to 48 primer pairs targeting at least one of the detected somatic variants. To ensure quality control, the personalized primer panel is complemented with a fixed primer panel comprising 21 common population-specific single nucleotide polymorphisms (SNPs). Subsequently, germline variants are identified by analyzing cfDNA from plasma with a buffy coat DNA control sample. The final step involves multiplex PCR amplification focusing on the genomic alterations selected as somatic variants.

The limitation of these promising bespoke assays is the considerable amount of tumor tissue needed to perform the assay. As an example, Signatera^TM^ requires a minimal tumor surface of 5 square millimeter (mm^2^), this can be a limitation considering the usual thickness of melanoma specimens. Furthermore, the cost of each test is very high to date, which decreases its cost-efficiency.

### 1.6. Clinical Applications of Current ctDNA Techniques

Current highly sensitive techniques like BEAMing and ddPCR are able to detect and quantify only a limited number of specific genomic alterations at a time. In melanoma, mutations in *BRAF* V600 (present in 40–60% of patients) or *NRAS* Q61 (found in about 25%) make these methods particularly effective for analyzing ctDNA in most cases (Table 2 and Figure 1, Figure 2 and Figure 3). However, in approximately 30% of melanoma cases that are wild-type (wt) for both *BRAF* and *NRAS*, ctDNA analysis is more challenging.

Mutations in the *TERT* promoter, specifically C250T and C228T, which appear in 30–85% of melanoma patients, offer an alternative marker for ctDNA monitoring [37,38]. Nevertheless, these mutations are in the promoter region, which is less stable in cfDNA because it lacks protective nucleosomes and histones, making it more prone to degradation.

Although these frequent mutations are valuable for tracking disease status, current techniques are limited as they cannot detect other mutations, including those that emerge during resistance development. To overcome this limitation, NGS approaches, such as WGS [53], WES [54], and targeted sequencing panels [55], provide a more comprehensive analysis. NGS methods can identify rare mutations within ctDNA at extremely low frequencies, enabling a broader understanding of tumor genomics, including the detection of new or co-occurring mutations [56].

To date, ddPCR and NGS technologies have proven to be complementary approaches for studying both common and rare genomic alterations, aiding in the detection of recurrence and monitoring treatment efficacy in melanoma patients. The advantage of NGS lies in the ability to capture much greater mutational information compared to ddPCR. However, ddPCR, is a quantitative technique, whereas NGS is only semiquantitative; considering standard and modified NGS techniques, emerging NGS bespoke assays are also quantitative (Table 2).

Nevertheless, quantifying ctDNA can be influenced by several physiopathological factors, such as inflammation, autoimmune diseases, pregnancy and physical exercise, or preanalytical factors primarily during blood collection [64].

### 1.7. Preanalytical Factors for Current ctDNA Techniques

There are several important preanalytical factors to consider in measuring ctDNA.

To maximize the potential to detect low abundance ctDNA, first, efforts should be made to collect as large a volume of blood as is feasible for analysis.

ctDNA is preferentially extracted from plasma rather than serum because the latter contains higher levels of non-tumoral cfDNA generated mostly from leukocyte lysis that occurs during the coagulation process [65]. The dilution of ctDNA by leukocyte cfDNA can adversely affect detection of the former, especially ctDNA harboring low MAFs [66].

Plasma volumes of 3–5 mL should be used to extract cfDNA, as these volumes will achieve a higher DNA yield than 1 or 2 mL of plasma, increasing the probability of detecting the mutation(s) of interest [67,68].

Furthermore, two key variables of preanalytical conditions of ctDNA obtention are the stability of the cfDNA and the potential for lysis of normal blood cells, leading to contamination with non-tumoral DNA. To limit these effects, ethylenediaminetetraacetic acid (EDTA)-coated blood collection tubes are most commonly used for cfDNA extraction. EDTA not only serves as an anticoagulant minimising the lysis of mononuclear cells but also inhibits DNase enzymes found in the blood, protecting cfDNA from degradation [69]. Other common anticoagulants such as heparin or citrate have been associated with lower cfDNA yields, perhaps because they do not inhibit DNases [70]. One limitation of EDTA-coated blood collection tubes is the commonly accepted need to process them quickly, generally within 6 h. After 6 h, hematopoietic cells will begin to lyse, releasing their DNA and RNA in the sample [71].

Alternatively, specialized cfDNA collection tubes that contain preservative reagents for leukocyte stabilization to prevent lysis can stabilize cfDNA and intact cells for up to 7 to 14 days at room temperature (18–25 °C), well known as streck cell-free DNA tubes [72,73], which are specifically designed for the preservation of cell-free nucleic acids. While these tubes can simplify the specimen collection and processing workflow, the cost of these tubes can be is almost 37 times more than EDTA blood collection tubes.

The centrifugation of blood samples is also a critical factor during the cfDNA extraction process. Plasma isolation from EDTA-coated tubes traditionally involves a single 10 min spin at approximately 1600× *g*, which eliminates most cells and platelets from the sample. However, this method is ineffective at eliminating all residual cells in plasma, and DNA from the remaining cells may significantly dilute the tumor DNA with cellular DNA [74].

Therefore, blood samples collected in EDTA tubes need to be processed by a double spin, 10 min at 1600× *g*, with the supernatant transferred to a second tube for a 10 min microcentrifugation at 16,000× *g* [74]. Importantly, the centrifugation protocols of specialized collection tubes may differ from EDTA tubes [68].

Nevertheless, as a further standardization step, cfDNA-specific guidelines developed by the National Cancer Institute (NCI) Biorepositories and Biospecimen Research Branch Biospecimen Evidence-Based Practices (BEBP) recommend shorter durations of room temperature storage for both EDTA (2–4 h) and preservative tubes (up to 3 days) prior to prescribed methods of plasma isolation and storage at −80 °C [75].

Importantly, cfDNA originates from multiple sources, including clonal hematopoiesis of indeterminate potential; this phenomenon, where “normal” blood cells carry somatic mutations in genes like *TP53* and *KRAS* [76], can result in false-positive clinical interpretations if these mutations are detected in cfDNA [77]. To mitigate this risk when analyzing mutant alleles as ctDNA, it is crucial to account for clonal hematopoiesis, which increases with age. This can be achieved by evaluating both peripheral blood mononuclear cells and cfDNA alongside the primary tumor sample [77], as performed by Signatera^TM^.

Finally, ctDNA results with quantitative techniques are reported with different units: ddPCR results can be reported as copies per milliliter (copies/mL) of plasma [78] as absolute measurement for mutant and wt copies, although some authors have shown their results as percent of reactions that are mutant [79], referring to VAF.

Contrastingly, a customized NGS panel known as Signatera^TM^ reports the results in MTM/mL of plasma, which takes into consideration the mean VAF of the 16 SNVs identified in the ctDNA of each patient and the total amount of cfDNA [80].

## 2. ctDNA Applications in Melanoma Patients

### 2.1. Resected High-Risk Melanoma: Stages IIB to IVD

#### 2.1.1. ddPCR

Detecting MRD could be especially beneficial for monitoring melanoma patients at high risk of recurrence following surgical removal of the tumor. However, since ctDNA levels typically correlate with tumor burden, it remains uncertain whether ctDNA detection is viable in early-stage melanoma (stage I–II), where the disease volume is generally low [81]. Further research is needed to assess the practicality of using ctDNA in these cases.

Bettegowda et al. [72] showed that early-stage cancers, encompassing stage I–II, often present with fewer than 10 copies of ctDNA per 5 mL of plasma. However, screening for ctDNA in stage III–IV melanoma patients after CLND or surgical resection of a metastasis could distinguish those patients with MRD needing further adjuvant therapies.

In one study, 161 stage II–III high-risk resected melanoma patients treated with adjuvant bevacizumab versus (vs.) placebo within the AVAST-M clinical trial were studied with ddPCR to detect *BRAF* and *NRAS* common mutations in plasma [82] (Table 3). *BRAF* and *NRAS* mutations in ctDNA, considered as a minimum of one copy of mutant/mL, were detected in 11% for *BRAF*-mutant and 14% for *NRAS*-mutant patient samples. Patients with detectable ctDNA after surgery had a significantly decreased disease-free interval (DFI), distant metastasis-free interval (DMFI) and OS regardless of trial arm. Also, postoperative ctDNA detection was also associated with worse DFI, DMFI and OS in multivariate analysis, independent of other clinical factors such as performance status (PS) and disease stage (*p* < 0.0001).

In another study [83], *BRAF* mutation was prospectively identified in 37/99 (37%) stage III melanoma patients, from a single institution through ddPCR assays in plasma samples. The detection of ctDNA at baseline and after resection was significantly correlated with worse DMFS and RFS, as confirmed by multivariate analysis after adjustment for disease stage and BRAF mutation status.

Moreover, the prognostic value of detecting ctDNA before surgery was confirmed in a separate cohort of patients with resectable stage III melanoma from a different institution. The post-operative detection of *BRAF*/*NRAS*/*TERT* mutations in ctDNA was a strong predictor of short RFS [83]. Also, regarding this issue, the presence of pre-operative ctDNA, assessed by ddPCR detecting *BRAF* and *NRAS* mutations, was detected in 34% of 119 patients with stage III melanoma who underwent CLND, and was significantly associated with tumor burden and worse MSS [81].

#### 2.1.2. NGS

Recently, techniques that analyze more than one genetic mutation at a time in ctDNA from melanoma patients have also been explored, such as emerging NGS techniques. Long et al. conducted a retrospective translational analysis using a tumor-guided, patient-specific panel of up to 200 variants, Invitae Personalized Cancer Monitoring™ test to monitor ctDNA levels in 1127 patients with stage IIIB-D/IV melanoma following resection treated with nivolumab plus ipilimumab vs. nivolumab alone in the phase III CheckMate 915 trial [84].

The overall prevalence of pre-treatment ctDNA was approximately 16% (95% CI: 14–18%) regardless of treatment type, which is consistent with findings from the AVAST translational study using ddPCR [82] for detection driver mutations *BRAF* and *NRAS*.

A trend towards a higher prevalence of ctDNA positivity was observed in more advanced sub-stages of stage III melanoma, with rates of 11% for IIIB (35/333), 18% for IIIC (110/596) and 41% for IIID (13/32). Pre-treatment ctDNA positivity was associated with an increased risk of recurrence, with a hazard ratio (HR) of 1.87 (95% confidence interval (CI): 1.48–2.36).

In addition, patients with ctDNA present from week 13 of therapy had a higher rate of relapse. Multivariate analysis showed that including ctDNA with clinical factors and LDH improved the prediction of RFS.

A recent study employed the tumor-informed assay RaDaR^TM^ (NeoGenomics, Inc., Fort Myers, FL, USA) to analyze ctDNA in 276 plasma samples collected prospectively from 66 resected stage II to IV melanoma patients undergoing definitive local treatment and some of the (neo)adjuvant treatment with anti-PD-1 monotherapy or with anti-CLTA-4 or BRAF/MEKi [85]. Plasma samples were obtained every 3–6 months over a period of up to two years. ctDNA was detected in at least one sample in 19 patients (29%), including six cases (9%) at the time point immediately after surgery. Detection of ctDNA at this was correlated with poorer OS (median OS of 22.7 months vs. NR, *p* = 0.01) and indicated a trend toward reduced RFS (median RFS of 15.7 months vs. NR, *p* = 0.07).

In 10 cases, ctDNA was detectable prior to disease relapse, with a median lead time of 128 days (range: 8–406 days). Notably, among patients receiving adjuvant systemic therapy, no statistically significant difference in RFS was observed between those with positive or negative ctDNA at the collection timepoint after surgery (median RFS 23.1 months vs. 50.8 months, *p* = 0.664).

Disease progression occurred in all patients with detectable ctDNA in post-surgery samples. However, 32% of patients (7 out of 22) who experienced recurrence had no detectable ctDNA in any plasma sample, including four individuals with distant progression. Therefore, in this case, although using a bespoke assay, the test was only able to detect recurrence in 68% of patients, even though the percentage of patients with a positive ctDNA test at least one time point was 29%, higher than previously reported for resected melanoma disease patients. However, these results should be interpreted with caution due to small sample size.

Due to low ctDNA concentrations in the localized setting and sequencing artifacts intrinsic to standard or optimized NGS targeted techniques, these are not the best options for monitoring recurrence. However, error-suppression strategies to reduce background noise and improve analytical specificity, like the ones that incorporate a tumor-guided, patient-specific panel test, such Signatera^TM^ or Invitae Personalized Cancer Monitoring™ used in Checkmate 915 translational ctDNA analysis, can allow NGS techniques to have at least the same detection rate as ddPCR in the adjuvant context, with the particularity that they can be useful in all melanoma patients, regardless of their mutational landscape. It is not the case for ddPCR used in the AVAST-M trial, the results of which could only be reproduced in patients carrying *NRAS* or *BRAF* driver mutations; however, the global percentage of detection was similar to that in Checkmate 915 using Signatera^TM^.

Tumor heterogeneity and clonal evolution might render single-gene mutations found in primary melanomas not very informative when tracking these in metastatic lesions. The ability to investigate the genomic profile beyond a single driver mutation is a key element in expanding the number of patients deriving benefit from the use of ctDNA to guide their clinical management. Bespoke tumor-informed ctDNA panels are highly sensitive assays to detect ctDNA, with the potential to improve risk stratification in the curative setting. This method consists of sequencing tumor tissue samples and selecting relevant mutations for a custom-built ctDNA assay.

However, although these bespoke techniques could be more sensitive, they still fail to detect a considerable percentage of recurrences, as shown with the RaDar^TM^ test, in which although the ctDNA-positive rate is higher than reported, only a 68% of recurrences have a positive ctDNA at some point.

Tumor-informed assays have, to date, a high economical cost and require a huge amount of tumoral tissue to compare with germline DNA, which can compromise its future implementation.

In contrast, ddPCR methods are far more economical, but the mutational profile must be known in advance, and >20% of melanoma patients do not harbor a known driver genomic alteration, which are known to be *BRAF* and *NRAS*-wt. Therefore, their ctDNA would be undetectable by ddPCR but not necessarily negative. Prospective randomized trials comparing both techniques help to clarify which strategy is more informative and cost-effective, and they might still be complementary in the resectable melanoma setting.

### 2.2. Unresectable Stage III to IV Melanoma

#### 2.2.1. Clinical Utility for Diagnostic

##### qPCR

The first studies evaluating potential applications of ctDNA in advanced melanoma patients were performed using fewer sensitive techniques, such qPCR. In one study, Idylla™, a qPCR system was used for the detection of *BRAF* and *NRAS* mutations in ctDNA of 19 patients with MM at baseline and during treatment [86] (Table 4). At baseline, 47% patients harbored a *BRAF* V600 and 15% harbored a *NRAS* mutation in ctDNA, which is in concordance with the matched tissue level of 84% determined by pyrosequencing. The presence of a plasma mutation at baseline did not correlate to OS or LDH; however, a correlation between ctDNA concentration and the presence of a plasmatic mutation (*p* < 0.005) was found. In another similar study, 46 patients with MM underwent plasma ctDNA and tissue *BRAF* mutation testing with qPCR. A *BRAF* mutation was found in 45.7% of ctDNA and 44.8% of tissue samples, with a concordance between both of 82.8%. Interestingly, in 18 patients, therapy with BRAF/MEKi was initiated based on the result of ctDNA without matched tissue [87]. The ORR of these patients was 77.8%, and median PFS was 6.0 months, comparable to pivotal trials data (Table 4). This study provides preliminary real-world data showing that treatment with BRAF/MEKi could be applied based on ctDNA results.

Similarly, in another study, *BRAF* V600 and *NRAS* mutation in plasma ctDNA from 56 mutated MM patients was determined using qPCR [88]. The higher tissue–plasma concordance was in melanoma patients, with a sum of diameters of ≥30 mm, ≥2 metastatic sites and elevated LDH. OS was significantly decreased in patients with a qPCR low quantification cycle (Cq) (*p* < 0.05). The standard qPCR used in these studies detected mutations if present in >1% of ctDNA but was limited by the presence of disproportionate amounts of wt alleles in the blood; therefore, it is not an adequate method for disease-monitoring or the initial molecular classification of MM patients (*BRAF*V600-mutant vs. wt).

##### ddPCR

As mentioned above, qPCR techniques for ctDNA applications have a low sensitivity, which translates in a high false negative rate. Thus, in this setting more sensitive techniques are needed such as ddPCR, which can detect mutant and wt DNA at <0.01%. In one study, ctDNA *BRAF*V600 mutant using ddPCR was detected at baseline in 71.8% of 32 patients MM patients. Overall, a significant correlation was observed between ctDNA copies/mL and metabolic tumor burden (MTB) (*p* < 0.001) measured by 18F-fluoro-D-glucose positron emission tomography/computed tomography (18F-FDG PET/CT) [89]. In this study, ctDNA was not detectable in patients with an MTB of ≤10 cm^3^. Patients with detectable ctDNA had a significantly shorter median PFS than patients with undetectable ctDNA (*p* < 0.05).

Wong et al. also showed that ctDNA levels correlate with qualitative analysis of MTB in MM patients. However, Wong reported results from MTB in mL, whilst the current study reported results in cubic centimetre (cm^3^) [90]. In another study, 19 patients with MM were tested for somatic mutations *BRAF*, *NRAS* and *TERT* in tumoral tissue, and afterwards, they were tracked in each patient’s plasma using ddPCR. Somatic mutations occurred in 89% of patients, of whom 41.2% had ctDNA detectable in their paired plasma, and ctDNA detection was associated with shorter PFS (*p* < 0.05) [91]. ctDNA was detected in 41.6% of *BRAF*-mutant cases (5/12), 50% of *NRAS*-mutant cases (1/2), and 35.7% of *TERT*-mutant cases (5/14).

Despite efforts to validate *TERT* promoter mutations in plasma using NGS panels [37,38], these mutations did not perform well in NGS custom panels due to the region’s high guanine-cytosine content. As a result, *TERT* promoter mutations are primarily analyzed using ddPCR. In one study [92], a ddPCR assay detected *TERT* promoter mutations in plasma samples of 22 patients with MM with a LoD of 0.17%, the concordance between plasma and tumor tissue was 68% (15/22). *TERT* detection has been reported to be associated with a bad prognosis; in this study, *TERT* ctDNA-negative patients had significantly longer PFS (*p* < 0.05), and co-existence with *BRAF* or *NRAS* mutations (observed in 55% of cases) in ctDNA was associated with poor DFS and MSS [37,38].

##### NGS

Several studies have reported the clinical utility of NGS techniques in advanced melanoma diagnosis and management. For instance, Calapre et al. identified somatic mutations in 20 out of 24 (83%) tumor biopsies from advanced melanoma patients using a custom sequencing panel targeting 30 melanoma-associated genes. In addition, matched plasma samples with detectable ctDNA revealed mutations in 16 out of 20 (70%) patients [93]. Notably, 89% (range 75–100%) of SNVs found in plasma by targeted sequencing were also detected in tumor tissue, suggesting that ctDNA can accurately reflect the tumor’s mutational landscape in melanoma patients.

In another study, a highly sensitive NGS panel targeting 54 cancer-related genes was used to analyze SNVs and copy number amplifications in ctDNA from patients with MM. SNVs were detected in 75% of patients, with an 85% concordance rate between tumor tissue and ctDNA for somatic SNVs at VAF of at least 0.5%, which increased to 100% concordance at a VAF of 1% [94]. Interestingly, complete concordance for hotspot driver mutations, such as *BRAF* V600 and *NRAS* Q61K, between tumor and ctDNA samples occurred even at low SNV burdens, with individual VAFs ranging from 0.2% to 28%. Furthermore, a higher SNV load (≥2 unique SNVs) and SNV burden (cumulative SNV VAF > 0.5%) were significantly associated with worse OS (*p* < 0.05), even after adjusting for clinicopathological variables in multivariate analysis.

Another study examined ctDNA samples from 74 treatment-naïve advanced melanoma patients using a custom NGS covering 30 genes and 123 amplicons including driver and targetable mutations [95]. At a recommended cfDNA input of 20 ng, the panel detected at least one cancer-associated mutation in 84% of patients, with a LoD for MAFs as low as 0.2%. However, consistent with findings from other NGS techniques, the high GC content of the *TERT* promoter region hindered its amplification, limiting the analysis of this critical region. Interestingly, the Guardant360^®^ CDx NGS panel, which uses hybridization-based probes, demonstrated improved performance in detecting *TERT* promoter mutations in ctDNA from various cancer types [55,56,57]. This success is likely attributed to specific probe designs employed during library preparation.

NGS panels offer a comprehensive approach to capturing the mutational landscape of advanced melanoma at diagnosis by enabling simultaneous analysis of multiple genes. However, technical challenges, such as the poor amplification of the *TERT* promoter region due to its high GC content, remain a limitation. This is significant because *TERT* promoter mutations are present in 30–85% of melanomas, including up to half of *BRAF* wt cases.

Another aspect that needs to be clarified with standard and optimized NGS techniques is that the results reported are qualitative, not quantitative, and expressed as % MAF or VAF in comparison to ddPCR or customized NGS assays such as Signatera^TM^, the results of which are quantitative and reported in copies/mL or MTM/mL depending on the technique used.

#### 2.2.2. Monitoring Disease Following Systemic Treatment Initiation

##### MM Treated with PD-1-Based Therapy

ddPCR

ctDNA has also been evaluated overall as a monitoring disease tool and for the detection of mechanisms of resistance. One study investigated the relationship between pre-treatment and the early on-treatment detection of *BRAF*, *NRAS* and *KIT* alterations in ctDNA using ddPCR and treatment outcome in 76 melanoma patients treated with nivolumab alone or in combination with ipilimumab [96]. ctDNA was detected in 53% of patients at baseline and was associated with higher LDH levels, more advanced stage and worse Eastern Cooperative Oncology Group (ECOG). Interestingly, patients with negative ctDNA at baseline who remained negative during treatment and those with positive ctDNA at baseline who became undetectable after 12 weeks of therapy showed a similar ORR, 77% vs. 72%, respectively. However, patients with positive ctDNA at baseline who remained positive after 12 weeks had an ORR of 6%, much worse than the other two groups. In addition, the first two groups had significantly longer PFS and OS compared to patients with positive ctDNA at baseline and after 12 weeks. Finally, ctDNA clearance at week 12 of treatment was confirmed in multivariate analyses to be a relevant predictive marker of response to ICI.

Importantly, ctDNA profiles in patients with predominant brain disease were not accurate predictors of ORR; this observation also appears in many studies reviewed below and will be further discussed. ctDNA analysis offers a potential method for distinguishing between pseudo-progression and true progression in melanoma patients. Pseudo-progression, defined as radiological signs of progression that are not confirmed as actual disease advancement upon further imaging, was investigated in a study by Lee JH and colleagues. They analyzed ctDNA levels of *BRAF* and *NRAS* mutations in plasma samples from 29 advanced melanoma patients who exhibited disease progression after 12 weeks of anti-PD-1 therapy [97]. The results showed that all patients with pseudo-progression had undetectable or more than a 10-fold decrease in ctDNA levels of *BRAF* and *NRAS* mutations compared to pre-treatment levels. In contrast, 90% of patients with confirmed progression had elevated or consistently high ctDNA levels 12 weeks after starting ICI therapy. There were even differences in one-year survival between patients with progression confirmed by RECIST criteria according to ctDNA evolution, with patients with favorable ctDNA at 12 weeks having a one-year OS of 82% vs. patients with rising or persistently elevated ctDNA at 12 weeks having a one-year OS of 39% (*p* < 0.05).

Also, in this area, 85 MM patients treated with anti-PD-1 monotherapy were followed for almost two years by determining *BRAF* V600E/K or *NRAS* Q61/G12/G13 in ctDNA (copies/mL plasma) [98]. Patients with undetectable ctDNA at baseline showed better OS and PFS compared to those with detectable ctDNA (25 to 233 copies/mL plasma), even in multivariate analysis adjusted for LDH level, ECOG and number of extracranial disease sites. However, patients with central nervous system (CNS)-only progression had undetectable cDNA at baseline and subsequent assessments. A positive correlation was also observed between ctDNA levels and the following: total metabolic tumor volume (MTV), assessed by 18F-FDG PET/CT; number of metastatic sites; and total tumor burden, assessed by the sum of the product of the two-dimensional diameters of each metastasis measured on imaging studies. Also, it is important to highlight from this study that, at baseline, before treatment started, patients with CR or patients with PR presented a median of 0 copies/mL of ctDNA vs. a median of 31 copies/mL ctDNA found in patients with SD or progression (*p* < 0.05).

Importantly, further in the ICI response monitoring setting, ctDNA *BRAF*, *NRAS* and *KIT* mutations were analyzed using ddPCR in melanoma patients with active brain metastases receiving anti-PD-1-based therapy, with longitudinal ctDNA plasma samples over the first 12 weeks of treatment (threshold 2.5 copies/mL plasma) [99]. Patients with undetectable ctDNA at baseline and during treatment had a longer median OS than patients with detectable ctDNA at baseline and during treatment (*p* < 0.01). Consistent with previous observations, ctDNA was not detected in any of the patients with exclusively intracranial disease. In contrast, ctDNA was detected in 64% of patients with extracranial disease; however, in the whole cohort, the detection rate was 52.7%. Detection was also associated with extracranial disease volume and treatment response (*p* < 0.01). CSF samples were not available in this study. As already observed, ctDNA cannot be relied upon as a surveillance tool for patients with only brain metastases; the blood–brain barrier may restrict the release of ctDNA into the circulation.

It is important to note that when the terms ctDNA ‘dynamics’, ‘kinetics’ or ‘variations’ are used, they refer to VAF or concentration (MTM/mL or copies/mL) measured between two time intervals, e.g., before the first and before subsequent treatment cycles. ctDNA kinetics were evaluated in another work driven by Herbreteau et al., in which plasmatic ctDNA *BRAF* or *NRAS* mutations were quantified by ddPCR at baseline and after 2–4 weeks of treatment in an exploratory cohort (*n* = 53) and a validation cohort (*n* = 49) of metastatic *BRAF* or *NRAS*-mutant melanoma treated with anti-PD-1 alone or in combination with anti-CTLA-4 [100]. In this study, an increase in ctDNA levels from week 2 to week 4, known as biological progression (bP), correlated with a lack of benefit from anti-PD-1 treatment. No patients in this group achieved 4 months of PFS, and only 13% achieved 1 year of OS. Patients without initial bP achieved a 4-month PFS in 78% of cases and 1 year of OS in 73% of patients.

However, interpretation of ctDNA kinetics during monitoring remains complex, as quantification by ddPCR can be imprecise when mutant copy numbers are low. At its limit of detection, the coefficient of variation (CV) of ddPCR can approach 100%. Therefore, the definition of bP based on ctDNA changes should be based on percentage deviations from a reference point rather than fixed thresholds, as highlighted in a previous study introducing the concept of bP [100].

A decrease in ctDNA is consistently linked to improved ORR, PFS, and overall survival OS. However, studies vary significantly regarding the thresholds (e.g., 20%, 50%, complete clearance) and the timing of the drop (e.g., after one infusion, 4–8 weeks) used to assess molecular response. Harmonising these strategies through standardized cutoff definitions and time points for baseline comparisons will be critical to design prospective clinical trials that allow ctDNA monitoring into clinical practice. Additionally, understanding intraday ctDNA variations and improving method reproducibility is essential to accurately identifying biological ctDNA changes [64].

On the other hand, the efficacy of ICIs depends on numerous factors, including transcriptionally regulated escape mechanisms, immune system composition, and tumor microenvironment interactions. Therefore, combining biomarkers to integrate multiple metrics will likely yield the most accurate predictions of tumor response to ICI [101]. Peripheral blood analysis, encompassing ctDNA, circulating cytokines, and T-cell population profiles, offers an ideal, comprehensive approach, as each have already shown promising results individually as biomarkers of clinical efficacy.

In this regard, Nabet et al. recently developed the DIREct-On score (Durable Immunotherapy Response Estimation) for predicting the response of non-small cell lung cancer (NSCLC) patients to ICI therapies. This method combines pretreatment biomarkers, ctDNA-normalized TMB, PD-L1 tissue expression, and circulating CD8+ T-cell fractions, with ctDNA levels after one ICI treatment cycle, analyzed using the CAPP-seq method [102]. The DIREct-On score outperformed individual metrics in clinical classification accuracy and prognostic value. In a multivariate Cox proportional model, it was the only factor independently associated with PFS, surpassing other variables such as age, ECOG, and line of therapy [102].

NGS

Monitoring disease by analyzing ctDNA kinetics in advanced melanoma patients with a multi-gene NGS panel or wide tumor genome evaluation with WES have also shown promising results. In a study, the peripheral blood ctDNA of 69 patients with diverse malignancies, including 10 patients with melanoma, who received anti-PD-1-based therapy were assessed with Guardant360^®^ CDx NGS panel, including 73 genes [103]. A higher number of VUS changes (>3) was significantly associated with improved PFS compared to a lower number of changes (≤3) (*p* < 0.05). In addition, clinical benefit, defined as SD for ≥6 months, PR, or CR, was observed in 45% of patients with VUS > 3 compared to 15% of patients with VUS ≤ 3 (*p* < 0.05).

Furthermore, even though tumor mutational burden (TMB, number of mutations per megabase [Mut/Mb]) has been proposed as an independent predictor of response to immunotherapy [104], the most promising results have been obtained in combination with plasmatic ctDNA evaluation. In a study focused on this topic [105], a tumor panel composed of 710 tumor-associated genes to reliably calculate TMB in liquid biopsies was performed in 35 MM patients treated with ipilimumab and nivolumab. TMB in the tumor biopsy was significantly higher (TMB > 23.1 Mut/Mb) in responders than in non-responders (TMB ≤ 23.1 Mut/Mb) before starting therapy. Furthermore, a >50% decrease in cfDNA concentration, measured by tumor-specific variant copies/mL of plasma, three weeks after treatment initiation, was significantly associated with response to ICI and improved OS. Finally, this same work revealed that the combination of high TMB (>23.1 Mut/Mb) and a >50% decrease in cfDNA concentration was a better predictor of response to ICI that both elements separately. The published cutoffs for high TMB in melanoma patients are similar to the ones obtained in this study [106].

Finally, an innovative technique with a bespoke ctDNA assay, Signatera^TM^, was performed on plasma samples at baseline and every three cycles, obtained from 94 patients with metastatic tumors from different origins, including 10 melanoma patients, treated with anti-PD-1 [62]. As previously described, peripheral blood and tumor tissue WES was performed to identify tumor-specific somatic mutations and differ them from germline mutations. For each patient, 16 clonal somatic mutations were selected for personalized ctDNA assay design with a ddPCR technique, which could be monitored during treatment course. This approach revealed the detection of ctDNA baseline levels down to 0.004% and 0.07 MTM/mL of plasma. Lower-than-median ctDNA baseline levels were associated with superior OS and PFS and clinical benefit rate (CBR). In comparison to absolute ctDNA at baseline, the relative change in ctDNA levels from baseline to C3 (∆ctDNAC3) displayed less variability across cancer types and was also associated with higher CBR and favorable OS and PFS. The choice of a bespoke ctDNA assay allowed the test to be applied to all patients with available WES data, whereas a fixed panel approach may not have identified mutations in all patients, and using 16 mutations may broaden the repertoire of potential candidates to test for ctDNA detection instead of selecting only a single mutation, especially in patients with tumors with unknown driver mutations.

To date there is no published data exploring ctDNA as biomarker in melanoma patients treated with nivolumab and relatlimab. A phase Ib trial evaluated neoadjuvant nivolumab or nivolumab-relatlimab in combination with chemoradiotherapy in 32 patients with resectable stage II/stage III gastroesophageal cancer. The primary endpoint was safety, while the secondary endpoint was feasibility [107]. Exploratory ctDNA analysis performed with Signatera^TM^ test showed that patients with undetectable ctDNA after neoadjuvant treatment, preoperatively, and postoperatively had significantly longer RFS and OS, regardless of treatment arm, nivolumab monotherapy or nivolumab in combination with relatlimab. Also, detectable ctDNA after neoadjuvant treatment correlated with residual tumor > 20% at the time of resection, and patients with undetectable ctDNA at that timepoint had a longer RFS compared to patients with detectable ctDNA.

Finally, a retrospective study is also worth mentioning, as it investigates the role of ctDNA monitoring using the Signatera^TM^ test in three different disease settings of melanoma patients treated with ICI [108]. In cohort A, consisting of 30 stage III patients receiving either adjuvant ICI or observation, those with persistent or increasing ctDNA levels after surgery had significantly worse DMFS compared to ctDNA-negative patients (HR, 10.77; *p* < 0.01). Similarly, in cohort B, which included 29 unresectable stage III/IV patients receiving ICI therapy, increasing ctDNA levels were also associated with worse PFS (HR, 22; *p* ≈ 0.006). Finally, in cohort C, which included 10 unresectable stage III/IV patients who were followed after planned completion of ICI therapy for advanced disease, those with persistently negative ctDNA remained progression-free during a median follow-up of 14.67 months, while ctDNA-positive patients progressed during the study period.

Finally, more recently, results from a prospective study evaluated ctDNA longitudinal monitoring in 87 melanoma patients receiving anti-PD-1-based treatments, including advanced disease (*n* = 65) and adjuvant approaches post-surgery (*n* = 22) with tumor-informed sequencing panels targeting up to 30 patient-specific mutations, from a targeted NGS analysis covering ≥ 700 genes. VAFs from plasma samples showed strong correlations with 18F-FDG PET/CT MTV (rho = 0.69), S100 protein levels (rho = 0.72), and LDH (rho = 0.54) [109]. A decline in VAFs between initial and follow-up measurements was linked to better PFS and OS in advanced melanoma patients (*p* = 0.008 and *p* < 0.001). In adjuvant cases, ctDNA was detected in 76.9% of patients who relapsed (10 of 13), up to 133 days before clinical or radiological progression, while no ctDNA was found in those without disease progression (*n* = 9).

The findings in this prospective study provide more evidence of the complementary roles of ctDNA and PET imaging in monitoring melanoma, and also in this study, ctDNA dynamics at treatment onset strongly predicted outcomes greater that baseline timepoints.

As shown, all these customized emerging NGS technique future applications could include increasing the number of monitored mutations to enhance sensitivity further and adopting broader genomic panels, even incorporating a previous WES or WGS analysis to expand mutation detection capabilities.

##### MM Treated with BRAF/MEK Inhibitors

qPCR

Several studies have shown that monitoring the levels of BRAF mutation ctDNA can be a useful tool for assessing the response to BRAF +/− MEKi in patients with *BRAF* V600-mutant melanoma. In general, a decrease in the levels of BRAF mutant ctDNA corresponds to a clinical or radiological response, while an increase in these levels suggests disease progression. However, it is important to note that these studies often involve small sample sizes, and further research is needed to establish the clinical utility of ctDNA monitoring in melanoma patients. In one study, allele-specific qPCR analysis for detecting *BRAF* V600 E/E2/D/K/R/M (Idylla) mutations on ctDNA was used to analyze plasma samples of patients with known *BRAF* V600-mutant melanoma treated with dabrafenib and trametinib in a phase II trial (*n* = 36 patients) [110]. At baseline, *BRAF* V600-mutant ctDNA was detected in 75% of patients and decreased rapidly upon the initiation of targeted therapy (*p* < 0.001), becoming undetectable in 60% of patients after 6 weeks of treatment. ctDNA dynamics at 8 days post-treatment correlated with disease control rate (DCR): in patients whose ctDNA decreased, the DCR was 75% compared to 18% in patients with stable or increasing ctDNA levels at this time. Also, patients with undetectable ctDNA levels after a median of 13 days of BRAFi therapy had longer PFS.

Regarding this same topic, another study used a method based on an allele-specific qPCR (Taqman) with higher sensitivity because of the presence of a peptide nucleic acid designed to inhibit amplification of the wt allele for detection and quantification of *BRAF* V600E, in ctDNA isolated from plasma and serum from 22 MM patients treated with BRAFi [111]. Median PFS according to *BRAF* V600E mutation detection in pre-treatment ctDNA was 3.6 months for positive patients vs. 13.4 months for negative patients (*p* < 0.05), and median OS was 7 vs. 21.8 months for positive vs. negative *BRAF* V600E testing at the same time point (*p* < 0.05). Also, patients with more advanced stages or ≥3 metastatic sites tended to have higher levels of *BRAF* V600E. No significant differences in response were observed according to the presence of the *BRAF* V600E mutation in pretreatment ctDNA.

More recently, the same authors published a multicentric study using the same method to analyze cfDNA from 66 advanced *BRAF* V600E/K melanoma patients treated with dabrafenib, in which *BRAF* mutation in pre-cfDNAs was associated significantly with tumor burden, PFS and OS (*p* < 0.05) [112]. Considering only patients with known *BRAF* V600E or *BRAF* V600K in tissue (*n* = 54), the concordance/sensitivity of the assay was 81.6% for *BRAF* V600E and 40% for *BRAF* V600K. Patients with ≥2 metastatic sites had a 16-fold higher *BRAF* V600 pre-cfDNA mutation load than those with <2 metastatic sites (*p* < 0.05). Similarly, patients with stage M1C had a significantly higher *BRAF* V600 pre-cfDNA mutation load than those with stage IVB or IVA (*p* < 0.05). In this study, patients were stratified according to ctDNA detection levels, which defined prognostic stratification subgroups in terms of PFS and OS. OS for patients with 0–10.5 picogram per microliter (pg/µL) *BRAF* V600 mutated genomes was 17.0 months, in comparison to 5.3 months for patients with more than 10.5 pg/µL *BRAF* V600 mutated genomes (*p* = 0.0002). Furthermore, PFS for patients with 0–10.5 pg/µL *BRAF* V600 mutated genomes was 8.8 months, compared with 3.6 months for patients with more than 10.5 pg/µL *BRAF* V600 mutated genomes (*p* = 0.0067).

Although, as already mentioned, qPCR has a lower sensitivity in ctDNA than more advanced techniques, these three studies show the usefulness of ctDNA using a qPCR in categorizing *BRAF*-mutant patients treated with BRAFi+/−MEKi in prognostic groups according to ctDNA and cfDNA *BRAF* V600 levels and its correlation with disease burden.

ddPCR and BEAMing

DdPCR is a highly sensitive technique that surpasses qPCR in analytical sensitivity. Its potential for monitoring *BRAF*-mutant melanoma patients’ responses to BRAF/MEKi and detecting resistance mechanisms has been extensively studied. In one investigation, plasma ctDNA *BRAF* V600E levels were measured using ddPCR in 8 controls and 20 patients with *BRAF* V600E-mutant advanced melanoma during BRAFi treatment. Sampling was performed at baseline, the first month of therapy, best response, and progression [52]. *BRAF* V600E mutation was detected at a fractional abundance of 0.005% in the wild-type gene, with a LoD established at 1 copy of mutant DNA/mL. Agreement between tumor tissue *BRAF* V600E status and plasma ctDNA *BRAF* V600E was 84.3%. Baseline ctDNA *BRAF* V600E correlated significantly with tumor burden (*p* < 0.05), and concentrations decreased significantly at the first month of therapy and at best response (*p* < 0.05). At disease progression, ctDNA *BRAF* V600E levels increased compared to levels at best response (*p* < 0.05). Importantly, lower baseline ctDNA *BRAF* V600E concentrations were significantly associated with improved outcomes, including longer OS and PFS, compared to higher baseline concentrations (27.7 vs. 8.6 months and 9 vs. 3 months, respectively, *p* < 0.05). A cutoff of 216 ctDNA mutant copies/mL was used to stratify patients for these analyses.

Additionally, in one study, *BRAF* V600 mutant melanoma patients treated with vemurafenib, another BRAFi, were assessed of ctDNA plasma concentration using a ddPCR targeting the *BRAF* V600E/K mutation [80]. At baseline, plasma ctDNA was detectable in 72% and the ctDNA concentration decreased in 88% of these patients on day 15 after vemurafenib initiation. Higher ctDNA concentration at baseline was associated with worse OS (*p* < 0.05), and interestingly, an inverse correlation between vemurafenib concentration and ctDNA concentrations was demonstrated (*p* < 0.05). Also, in another study on this topic, plasma ctDNA *BRAF* V600E/K mutations were analyzed using ddPCR in 19 melanoma patients treated with BRAF/MEKi, and ddPCR negativity was confirmed with ultra-deep sequencing [113]. OS was significantly worse for patients with elevated LDH (*p* < 0.05) or detectable ctDNA (*p* < 0.05) at the start of targeted therapy. Importantly, in patients with progression disease confined to the brain, ctDNA results did not correlate well with OS. Also, as seen previously, ctDNA determination using ddPCR has not shown its utility in monitoring patients with only intracranial disease, probably due to low shedding in peripheral blood when the disease is confined to the brain.

Importantly, disease site location influences ctDNA release, and patients with visceral, bone, or lymph node involvement exhibited higher levels of ctDNA, which is out of keeping with the metabolic disease burden as assessed by 18F-FDG-PET/CT, whilst those with extensive subcutaneous disease or with brain metastases showed consistently low levels of ctDNA despite measurable disease.

The limited utility of ctDNA as a biomarker of intracranial response suggests that peripheral blood ctDNA analysis and clinical imaging are complementary rather than interchangeable methods. In MM, intracranial disease control has not been consistently correlated with ctDNA variations, quantifiable units or fractions across multiple studies. Conducting well-designed studies that simultaneously assess tumor response using both methods could provide valuable insights into their complementary roles and help develop more accurate models for predicting clinical outcomes [114]. In addition, the use of highly sensitive custom NGS assays for ctDNA analysis or CSF as an alternative source represents a promising avenue for patients with primarily intracranial disease, as discussed further in Section 2.3. However, when considering the latter strategy, it should be borne in mind that CSF ctDNA analysis is a much more invasive procedure.

Recently, ddPCR assays were employed to measure *BRAF*V600-mutant ctDNA in both pretreatment and on-treatment plasma samples from patients in two clinical trials [115]. The COMBI-d trial was a double-blind, randomized phase III study comparing dabrafenib plus trametinib to dabrafenib with a placebo in previously untreated patients with *BRAF*V600-mutant MM. Meanwhile, the COMBI-MB trial was an open-label, non-randomized phase II study that assessed the efficacy of dabrafenib combined with trametinib in patients with *BRAF*V600-mutant MM and CNS metastases. In the COMBI-d and COMBI-MB studies, 320 (93%) of 345 patients and 34 (89%) of 38 patients, respectively, had a *BRAF* V600 mutation detected in baseline ctDNA before treatment initiation, and a high baseline *BRAF* V600 ctDNA concentration was associated with worse OS and PFS (*p* < 0.05), with an optimized cut-off of 64 copies/mL remaining statistically significant for OS, independent of clinical factors (*p* < 0.05). Baseline ctDNA concentration correlated with baseline sum of extracranial (*p* < 0.05) but not intracranial metastatic lesion diameters, as observed in other studies (*p* > 0.05), and ctDNA negativisation (including all longitudinal samples during treatment) was associated with best extracranial overall response (BOR) (*p* > 0.05) but not intracranial BOR (*p* > 0.05).

More data from clinical trials are also worth mentioned. A pooled analysis evaluating 732 MM *BRAF* V600 MM patients treated with BRAF+/−MEKi inside four clinical studies, including BREAK-2, BREAK-3, BREAK-MB and METRIC [47]. Baseline ctDNA was detected in 76% of *BRAF* V600E mutant patients and 81% of *BRAF* V600K mutant patients. In this pooled analysis, patients with undetectable *BRAF* V600 ctDNA at baseline analyzed by the BEAMing assay had longer PFS and OS and a higher response rate. ctDNA mutation fraction was positively correlated with baseline sum of lesions diameter and LDH across studies and a worse ECOG was associated with a higher V600E/K mutation fraction in circulating plasma. As observed in other studies, patients with visceral disease at baseline tended to have higher mutation fractions for both V600E and V600K.

Likewise, another potential application of ctDNA in this context is detecting the emergence of resistance mechanisms. *NRAS* mutations, particularly *NRAS* Q61K/R, are a commonly observed acquired resistance mechanism in patients with *BRAF*-mutant melanoma treated with BRAF+/−MEK inhibitors. The detection of *NRAS* mutations in ctDNA can be used to monitor the emergence of resistance and potentially guide treatment decisions. In one study, *NRAS* mutations were detected in 43% of melanoma patients on treatment not present at baseline, receiving vemurafenib, dabrafenib or dabrafenib plus trametinib, before clinical and radiological progression, in their plasma ctDNA samples, suggesting the potential of ctDNA as an early predictor of resistance mechanisms [42]. However, *NRAS* mutation analysis in tumoral tissue at the time of ctDNA findings was not performed.

Another study identified acquired *NRAS* and *PIK3CA* mutations in cfDNA samples obtained from melanoma patients who had progressed on BRAF/MEKi [116]. WES and ddPCR were used to analyze the cfDNA samples and identify these mutations. Furthermore, *BRAF* gene amplification is another mechanism of acquired resistance to BRAF/MEK inhibitors that can lead to treatment failure. In this case, monitoring the ctDNA levels of mutant *BRAF* may not accurately reflect the disease status, as the increase in ctDNA levels may be due to the amplification of the wt *BRAF* gene rather than the mutant allele. Therefore, it is important to use a combination of different genomic techniques, such as WES or targeted sequencing, to detect various types of genetic alterations that may lead to resistance, in addition to monitoring ctDNA levels [116].

Lastly, another issue evaluated has been residual disease in patients with durable CR as a guarantee to discontinue treatment with BRAF/MEKi; for those cases, it is not clear whether it is safe to cease therapy. In one study, 13 patients treated with BRAF/MEKi who ceased therapy after prolonged CR (median 34 months, range 20–74) were monitored retrospectively using ddPCR in ctDNA in longitudinal plasma samples to detect *BRAF* V600E/K mutations [117]. Levels were undetectable in 11/13 cases after cessation and eventually became detectable in 2/3 cases with disease recurrence but remained undetectable in 1 patient with only intracranial progression. Also, on this topic, another retrospective analysis conducted at a single institution analyzed 24 patients with *BRAF*-mutant MM treated with a BRAF/MEKi, which interrupted treatment due to cumulative toxicity after achieving CR or long-lasting PR (>12 months) [118]. CR and PR were achieved in 71% and 29% of patients, respectively. At a median follow-up of 37.8 months (range 33.7–41.9) after treatment discontinuation, patients who achieved a CR and had undetectable ctDNA at discontinuation as measured by ddPCR showed significantly improved PFS compared to patients with either radiologically detectable residual disease or positive ctDNA (*p* < 0.05). Thus, analysis of ctDNA in patients who are stopping BRAFi+/−MEKi may reveal which patients have minimal residual disease and would benefit from ongoing targeted treatment.

As demonstrated with qPCR, ddPCR can effectively monitor disease progression in *BRAF*-mutant patients undergoing treatment with BRAF/MEKi. While ctDNA is detectable at baseline, the persistence of positive ctDNA during treatment and follow-up is a poor prognostic indicator, correlating with extracranial disease response. However, patients with exclusively CNS disease are not well tracked with plasma ctDNA, as its levels do not adequately reflect intracranial tumor activity. In addition to tracking disease burden, ddPCR can identify mechanisms of resistance to BRAF/MEKi therapy, such as *BRAF* gene amplification or the emergence of *NRAS* mutations, providing crucial insights into therapeutic resistance.

Finally, the BEAMing technique has shown potential as a prognostic and predictive biomarker in *BRAF*-mutant patients treated with BRAF/MEKi. However, despite its early promise, BEAMing has not gained traction in subsequent investigation, limiting its clinical adoption.

Also, in contrast to ddPCR, which reports results as mutant and wild-type copies/mL of a single mutation, BEAMing results are reported as mutation fraction, which refers to the relative abundance of mutant to wild-type of a single mutation in cfDNA, ctDNA or plasma.

NGS

NGS offers a broader scope compared single-gene techniques; however, for monitoring *BRAF* V600-mutant melanoma patients treated with BRAF/MEKi, its use may not be necessary. Since the driver mutation is well-known, single-gene methods can sufficiently track disease progression and response to therapy.

Nonetheless, a retrospective study [119] utilized blood samples from melanoma patients collected at diverse timepoints before or after treatment to evaluate correlation between mutations identified in biopsies and ctDNA using a NGS approach. Noteworthy, ctDNA sequencing conducted after targeted treatment in melanoma failed to detect mutations in most patients, likely reflecting treatment response.

However, the possibility of limited sensitivity cannot be excluded, raising concerns about the utility of this method for monitoring treatment efficacy. Additionally, the study found that ctDNA VAF was significantly correlated with tumor VAF (*p* < 0.05), when samples were collected within a year of the biopsy. In contrast, samples taken more than one year after the biopsy showed no correlation, suggesting that temporal factors influence the concordance between ctDNA and tumor profiles.

##### MM Treated with ICI or BRAF/MEKi

ddPCR

In a study already mentioned above, 73% of 48 MM patients were found to have positive ctDNA for *BRAF* and *NRAS* mutations by ddPCR prior to treatment with anti-PD-1 +/− anti-CTLA-4 or BRAF/MEKi [42]. Lower baseline ctDNA levels were significantly associated with treatment response and PFS prolongation, regardless of treatment type. Similarly, lower ctDNA levels at eight weeks post-treatment were associated with a better response. However, this decrease was more apparent amongst patients treated with and responding to targeted therapy (*p* < 0.05) compared to anti-PD-1 +/− anti-CTLA-4. Also, a significant correlation was found between the concentration of ctDNA and LDH levels (*p* < 0.05). In this study, as already mentioned, *NRAS* mutations were detected in three of seven progressing on BRAF/MEKi, with a ctDNA rebound and circulating mutant *NRAS* preceding radiological progression.

In this same context, a prospective study, including 43 MM patients, *BRAF*, *NRAS* and *KIT* mutations in ctDNA and basal concentration were analyzed using ddPCR and correlated with basal and changes in tumor burden and OS (*p* < 0.05) [120]. A cut off value of ≥89 pg/µL of ctDNA identified patients with shorter OS (*p* < 0.05), and the significance was maintained when compared with LDH in a multivariate analysis. Contrastingly, no correlation was found between plasma total ctDNA concentration and *BRAF*, *NRAS* or *KIT* mutations.

One study involving 96 patients tested for *BRAF* V600E, *NRAS* Q61, and *TERT* promoter mutations using ddPCR assays found that elevated baseline ctDNA levels were an independent predictor of disease progression, outperforming traditional markers like S100 and LDH (*p* < 0.05) [121]. Elevated *BRAF* V600E ctDNA levels were linked to shorter PFS (*p* < 0.001), while *NRAS* Q61 mutations were detected both at baseline and during therapy probably due to resistance mechanisms, correlating with shorter PFS and OS (*p* < 0.05).

In another study of 110 patients, low ctDNA levels before first-line treatment initiation ICI or BRAF/MEKi were associated with significantly longer PFS (*p* < 0.0001) [122]. However, this correlation was not observed for patients starting second-line ICIs after failing BRAF/MEKi, a finding that requires further confirmation. Additionally, combining anti-PD-1 with anti-CTLA-4 in patients with high baseline ctDNA levels showed a non-significant trend toward better PFS and OS than anti-PD-1 monotherapy.

A further study of 142 advanced melanoma patients (70 receiving BRAF/MEKi and 72 receiving ICIs) found that baseline ctDNA was detectable in 56% of cases, with declining ctDNA within 12 weeks strongly associated with better outcomes (*p* < 0.001 for PFS, *p* < 0.05 for OS) [123]. ctDNA decline was slower in patients treated with ICI compared to BRAF/MEKi. In this study, most patients who responded to ICI (67%) had detectable ctDNA levels at first follow-up at 3 to 6 weeks and only had a significant drop to undetectable levels on their second follow-up at 12–18 weeks (*p* < 0.05).

A meta-analysis of nine studies, encompassing 617 advanced melanoma patients, demonstrated that detectable baseline ctDNA was strongly correlated with poor OS and PFS (*p* < 0.001 for both) [124]. During treatment, whether with anti-PD-1-based immunotherapy or targeted therapy with BRAF/MEKi, ctDNA positivity remained a marker of worse prognosis for both PFS and OS (*p* < 0.001). No significant heterogeneity in methodology or patients’ characteristics was observed among the studies included in the meta-analysis, for either OS and PFS outcomes. As a result, a random-effects model was ultimately applied to account for any residual variability.

Finally, in a study of 25 MM patients receiving BRAF/MEKi or ICI, serial ctDNA analysis was performed alongside 18F-FDG-PET/CT imaging [125]. Plasma samples were analyzed using NGS and ddPCR to track mutant *BRAF*, *NRAS* and *TERT* mutations ctDNA levels, which closely reflected changes in metabolic disease burden during treatment. Patients who showed an early decline in ctDNA levels (1 to 4 weeks) showed improved PFS compared to those whose ctDNA levels increased or remained stable. However, changes in the volume of subcutaneous and cerebral disease sites during treatment were not well represented in the evolution of ctDNA levels in comparison to other disease sites. In addition, changes in *TERT* mutant ctDNA correlated closely with treatment response.

All these studies mentioned show that positive ctDNA analyzed by ddPCR detecting *BRAF*, *NRAS*, *KIT* and *TERT* at baseline correlates with response, PFS and OS, regardless of treatment type. However, low baseline ctDNA levels in patients commencing ICI as second-line therapy after failing therapy with BRAF/MEKi was not a predictor of longer PFS in one study.

There is controversy regarding whether the anti-PD-1 plus anti-CTLA-4 combination is more effective in patients with higher ctDNA at baseline, and studies exploring this issue have controversial results.

Interestingly, a slower decline in ctDNA levels was observed in patients receiving immunotherapy compared to those treated with BRAF/MEKi, suggesting that optimal timepoints for ctDNA monitoring may vary depending on the treatment strategy [123].

A study found that most patients who responded to immunotherapy initially had detectable ctDNA levels at their first follow-up. However, by the second follow-up, 12–18 weeks later, these patients showed a significant drop in ctDNA levels to undetectable levels.

A significant advancement in the clinical utility of ctDNA for melanoma patients is its use in guiding treatment decisions based on ctDNA kinetics, currently being evaluated in a prospective clinical trial. The phase II Circulating Tumor DNA Guided Switch (CAcTUS) trial (NCT03808441) focuses on MM patients with *BRAF* V600 mutations. Patients receiving dabrafenib plus trametinib are switched to nivolumab plus ipilimumab combination as second-line therapy based on *BRAF*-mutant ctDNA levels. The trial aims to enhance immunotherapy response by using prior targeted therapy and switching treatments based on ctDNA response, rather than waiting for resistance to develop. This trial consists of two arms, one standard arm in which no intervention is undertaken, patients start dabrafenib plus trametinib combination and switch to nivolumab and ipilimumab at first progression, and interventional arm in which patients also start dabrafenib plus trametinib combination and switch to nivolumab plus ipilimumab when there is evidence of response defined by a decrease in ctDNA *BRAF*-mutant VAF of ≥80% measured by ddPCR.

NGS

The NGS technique, including the WES approach, has also been explored as a biomarker in studies with advanced melanoma patients treated with BRAF/MEKi or ICI; however, neither of the studies reviewed found a relationship between ctDNA levels and treatment efficacy or prognosis, probably due to small sample sizes. In one study, plasma cfDNA was extracted from 25 advanced-stage melanoma patients and sequenced using a 61-gene panel. Most patients (84.0%) had received ≥ 1 previous therapies, and seven patients received ≥ 3 previous treatments, including ChT, BRAF/MEKi or ICI [125]. One or more mutations were detected in 48% of patients, and this proportion did not vary significantly for patients on or off therapy at the time of blood draw. Most frequent mutations detected in *BRAF*, *NRAS*, and *KIT*, were consistent with NGS analysis of TCGA, with VAF ranging from 1.1 to 63.2% (median 29.1%). Among patients with tissue NGS, concordance with plasma findings was 81.8%. ctDNA concentrations were positively correlated with tumor burden (*p* < 0.05), with no mutations found in the plasma of any patients, with a total tumor burden of ≤4.9 cm. No data about treatment response correlation were described in the article.

In another study, Oncomine™ Pan-Cancer Cell-Free Assay, a 52-gene panel, was performed on a tumor and was compared to its application on ctDNA plasma samples from 22 advanced melanoma patients before first-line treatment (BRAF/MEKi or ICI) [126]. The analysis showed a concordance of 91% between tissue and plasma NGS results for *BRAF* mutations and 28% for non-*BRAF* mutations. The Ion Torrent HD method, which uses dual-molecular barcoding, enabled the detection of very rare alterations down to VAF greater than 0.1%. All the patients with a cfDNA concentration above 50 ng/mL had metastasis in at least three sites, and cfDNA level correlated positively with the LDH concentration. However, no statistically significant correlation of the cfDNA level with PFS was found in the cohort, which may be due to the small sample size.

The concordance of NGS from cfDNA/ctDNA and tissue is above 80% and even higher when it comes to *BRAF* V600 in comparison to non-*BRAF* mutations. However, in studies evaluating NGS in patients treated with ICI or BRAF/MEKi, no correlation was found with prognosis or treatment efficacy, probably due to small sample sizes. Contrastingly, ctDNA concentrations in these studies correlated with tumor burden and number of disease sites.

##### Other Treatments

qPCR and ddPCR

Also, ctDNA potential as a predictive biomarker of response has been evaluated with other treatment strategies beyond BRAF/MEKi and ICI in melanoma patients. To begin with, the autologous transfer of Tumor-Infiltrating Lymphocytes (TILs) is a form of immunotherapy that involves the isolation of TILs from a patient’s own tumor, which are then expanded ex vivo and infused back into the patient to target and eliminate cancer cells. This treatment approach has shown promising results in melanoma patients, with some patients achieving long-term responses, including CR, even after a year or more following treatment [127]. In one study, *BRAF* V600E cfDNA levels were analyzed by a competitive allele specific TaqMan PCR (castPCR™) assay in serum samples from 48 patients who received TILs at the NCI. Patients that developed an early ctDNA peak of serum *BRAF* V600E ctDNA and cleared it within the first month after TILs transfer were highly likely to achieve a CR over the next 1–2 years.

Alternatively, in another study with 26 MM patients treated with bevacizumab, those with >1% *BRAF*/*NRAS* ctDNA assessed by ddPCR at baseline and during treatment had significantly decreased PFS and OS [128]. On the contrary, patients with ≤1% *BRAF*/*NRAS* ctDNA and normal LDH levels had an increased response to treatment; however, *BRAF*/*NRAS* ctDNA was a stronger predictor of response in comparison to LDH (*p* < 0.05). In this same study, it was also observed that ctDNA *BRAF* V600D/E/K and *NRAS* G12V/Q61K/L/R determination were better biomarkers for response than *TERT* promoter mutations.

Finally, a longitudinal ctDNA analysis was performed using Signatera^TM^ customized test within a phase II clinical trial evaluating tebentafusp in metastatic uveal melanoma patients. ctDNA clearance or early treatment reduction between baseline and weeks 5, 9 or 25 was strongly associated with survival [129]. Of the 127 patients in the trial, 118 (93%) had evaluable serum samples, with most (109/118; 92%) found to have detectable ctDNA at any timepoint up to and including week 9 (baseline, week 5, week 9). A total of 84% had detectable ctDNA at baseline, with 94 of them showing mutations in one or more uveal melanoma-related genes (GNAQ, GNA11, SF3B1, PLCB4, CYSLTR2) at a VAF > 0.3, similar to reported data in other melanoma subtypes. Baseline ctDNA levels, measured as MTM/mL, were strongly correlated with tumor burden, as defined by the RECIST sum of the longest target lesion diameters and baseline LDH levels [129].

It seems that, as with other treatment strategies such as TILs or bevacizumab within clinical trials, higher ctDNA *BRAF* V600 or even *NRAS* at baseline detected by qPCR or ddPCR is also prognostic and predictive of response in case of ctDNA clearance. Also, in other rare melanoma types, such as uveal, ctDNA could help identify those patients benefiting from tebentafusp treatment.

**Table 3 ijms-26-00861-t003:** Studies of ctDNA in localized stages (I–III) and resected stage IV melanoma patients.

Author Publication Date [Ref.]	N. Pts	Stage	FUP	Treatment	Age Median (Range)	Sex (M/F) (%)	Mutation	Method	Analytical Sensitivity (LoD)	Detection Rate (%)	Associated Variables	Cut Off: Positive Value or Prognostic
Lee 2018 [82]	161	II III	5-year	adjuvant bevacizumab vs. placebo (AVAST-M trial)	52 y (19–87)	48/52	*BRAF*, *NRAS*	ddPCR	0.01%	12	OS, DFI, DMFI	Positive value: ≥1 copy/mL
Tan 2019 [83]	99	III	20 mo	anti-PD-1 adjuvant	57 y (22–93)	71/29	*BRAF*, *NRAS*, *TERT*	ddPCR	NR	37	RFS, DMFS	Positive value: ≥1 copy/mL
Lee 2019 [81]	119	III	26 mo	NR	64 y (20–90)	66/34	*BRAF*, *NRAS*	ddPCR	NR	34	MSS	Positive value: ≥1 positive droplets
Long 2022 [84]	1127	IIIB-D/IV	NR	adjuvant nivo + ipi vs. nivo	56 y (45–67)	57.5/42.5	tumor specific alterations	WES PCR	NR	16	RFS, DMFS	NR
Genta 2024 [85]	66	II–IV	39 mo	(neo) adjuvant anti-PD-1 +/− anti-CTLA-4 or BRAF/MEKi	65 y (27–87)	29/71	Tumor specific alterations	WES and personalized ddPCR-NGS	NR	29	OS, RFS	ctDNA+: pre-set threshold defined in assays’ analytical development
Eroglu 2023 [108]	30 (cohort A)	III	19.6 mo	adjuvant nivo	72 y (21–90)	53/47	tumor specific alterations	WES and personalized ddPCR-NGS	0.004%	17	MRD, DMFS	Positive value: 0.07 MTM

Abbreviations: F, female; FUP, follow-up; ipi, ipilimumab; M, male; mo, months, nivo, nivolumab; pts, patients; Ref., reference.

**Table 4 ijms-26-00861-t004:** Studies of ctDNA in advanced stages (III irresectable and IV) melanoma patients.

Author Publication Date [Ref.]	N. Pts	Stage	FUP	Treatment	Age Median (Range)	Sex (M/F) (%)	Mutation	Method	Analytical Sensitivity (LoD)	Detection Rate (%)	Associated Variables	Cut Off: Positive Value or Prognostic
Long-mira 2018 [86]	19	IV	NR	BRAF/MEKi or anti-PD-1 +/− anti-CTLA-4 or ChT	61.63 y (43–78)	84/16	*BRAF* *NRAS*	allele-specific qPCR	0.1%	88	ctDNA concentration and presence of *BRAF/NRAS* mutation	NR
Sobczuk 2022 [87]	46	III IV	≥12 mo	BRAF/MEKi	NR	54/46	*BRAF*	qPCR	>1%	72.4	NR	NR
Giunta 2022 [88]	56	III IV	18.7 mo	BRAF/MEKi or anti-PD-1 +/− anti-CTLA-4	62 y (34–86)	53.3/46.7	*BRAF* *NRAS*	qPCR	NR	60	tumor burden, OS	NR
McEvoy 2018 [89]	32	IV	64.4 w	BRAF/MEKi or anti-PD-1 +/− anti-CTLA-4	57 y (25–83)	62.5/37.5	*BRAF*	ddPCR	NR	71.8	MTB, PFS	NR
Marcynski 2020 [91]	19	III IV	130 d	NR	<61 y	35.3/64.7	*BRAF NRAS TERT*	ddPCR	0.13–0.37%	41.2	PFS	NR
McEvoy 2017 [92]	22	IV	NR	treatment naïve	51 y (24–81)	NR	*TERT*	ddPCR	0.17%	68	PFS	NR
Calapre 2019 [93]	24	IV	NR	anti-PD-1 +/− anti-CTLA-4	51–70 y	70/21	*BRAF* *NRAS* *TERT*	ddPCR	NR	70	NR	NR
Lee 2017 [96]	76	IV	17.5	nivo or nivo + ipi	65 y	60/40	*BRAF NRAS* *KIT*	ddPCR	NR	53	PFS, OS	Positive value: >2 positive droplets
Lee 2018 [97]	29	IV	84 w	anti-PD-1 +/− anti-CTLA-4	65 y	62/38	*BRAF* *NRAS*	ddPCR	NR	93.1	PFS, OS	NR
Seremet 2019 [98]	85	III IV	84 w	anti-PD-1	57 y (27–82)	43.5/56.5	*BRAF* *NRAS*	ddPCR	0.01%	44.4	PFS, OS, TMTV	Positive value: >2 mutant copies per PCR Prognostic stratification: >500 copies/mL
Lee 2020 [99]	72	IVD	35.6 mo	anti-PD-1 +/− anti-CTLA-4	65 y	68/32	*BRAF NRAS* *KIT*	ddPCR	NR	52.7	response PFS, OS, tumor burden	Positive value: >2.5 mutant copies/mL
Herbreteau 2020 [100]	53 (exploratori cohort) 49 (validation cohort)	IIIC- IV	NR	anti-PD-1 +/− anti-CTLA-4 or BRAF/MEKi	62 y (52.5–72.4)	54.4/45.5	*BRAF* *NRAS*	ddPCR	NR	50	PFS, OS	Positive value: >8 mutant copies/mL
Schreuer 2016 [110]	36	IV	NR	dabrafenib or dabrafenib + trametinib or vemurafenib	52 y	33/67	*BRAF*	allele specific qPCR	NR	75	DCR, PFS	NR
Gonzalez-Cao 2015 [111]	22	IV	NR	BRAFi	62 y (35–83)	63/27	*BRAF*	allele specific qPCR	0.005%	57.7	PFS, OS	Positive value: *BRAFV600* allele amplified in 2 of 4 quadriplicates
Gonzalez-Cao 2018 [112]	66	IV	NR	BRAF/MEKi or ipior ChT	58 y (28–44)	48/52	*BRAF*	allele specific qPCR	0.005%	66.7	tumor burden, PFS, OS	Prognostic stratification: High > 10.5 pg/µL Low–indetectable 0–10.5 pg/µL
Sanmamed 2015 [52]	20	IV	NR	BRAFi	50 y	65/35	*BRAF*	ddPCR	0.005%	84.3	tumor burden, PFS, OS	Positive value: ≥1 mutant copies/mL Prognostic stratification: >216 copies/mL
Forschner 2020 [113]	19	IV	NR	BRAF/MEKi	51 y (32–79)	42/58	*BRAF*	ddPCR	NR	68	OS	NR
Santiago-walker 2016 [47]	732	IV	NR	dabrafenib (BREAK-2 trial) dabrafenib vs. DTIC (BREAK-3 trial) dabrafenib (BREAK-MB trial) trametinib vs. ChT (METRIC trial)	NR	NR	*BRAF*	BEAMing	0.01%	81	ORR, PFS, OS	NR
Syeda 2021 [115]	383	IIIC- IV	20 mo	Dabrafenib + trametinib (COMBI-d, COMBI-MB trials)	56 y (45–65)	53/47	*BRAF*	ddPCR	0.019–0.022%	89–93	PFS, OS, BOR	Positive value: 0.28 mutant copies/mL if *BRAF* V600E and 0.34 mutant copies/mL if *BRAF*V600K Prognostic stratification: >64 copies/mL
Gray 2015 [42]	48	IV	NR	Vemurafenib or dabrafenibdab + trametinib or pemor nivo + ipi	NR	NR	*BRAF* *NRAS*	ddPCR	0.01%	65	ORR, PFS	Positive value: ≥1 mutant copies/mL Prognostic stratification: ≥10 copies/mL
Warburton 2020 [117]	13	IV	57 mo	BRAF+/−MEKi at discontinuation	61 y (38–71)	54/46	*BRAF*	ddPCR	NR	15	MRD	Positive value: ≥1 positive triplicate
Di guardo 2021 [118]	24	IV	37.8 mo	BRAF+/−MEKi at discontinuation	56 y (43–63)	50/50	*BRAF*	ddPCR	0.1%	NR	PFS after treatment discontinuation	NR
Valpione 2018 [120]	43	IV	11.9 mo	Ipi or BRAF/MEKi anti-PD-1 or DTIC	58.1 y (18–85.1)	58/42	*BRAF NRAS* *KIT*	ddPCR	NR	70	tumor burden, OS	Prognostic stratification: ≥89 pg/μL
Varaljai 2019 [121]	96	III IV	NR	BRAF/MEKi or anti-PD-1 +/− anti-CTLA-4	NR	NR	*BRAF NRAS* *TERT*	ddPCR	NR	NR	response, PFS, OS	NR
Marsavela 2020 [122]	110	IV	95 w	BRAF/MEKi or anti-PD-1 +/− anti-CTLA-4	65 y	65/35	*BRAF* *NRAS*	ddPCR	NR	NR	PFS (only 1 L ICI patients) OS	Prognostic stratification: ≤20 copies/mL
Marsavela 2020 [123]	142	IV	113 w	anti-PD-1 +/− anti-CTLA-4 or BRAF/MEKi	NR	NR	*BRAF* *NRAS*	ddPCR	NR	65	response, PFS, OS	NR
Wong 2017 [90]	52	IV	391 days	BRAF/MEKi or immunotherapy	61 y (24–83)	NR	*BRAF NRAS TERT*	ddPCR	0.1%	77	tumor burden, PFS	NR
Xi 2016 [127]	48	IV	NR	TILs	NR	NR	*BRAF*	allele-specific qPCR	0.05%	NR	CR after 1–2 years	NR
Forthum 2019 [128]	26	IV	NR	bevacizumab	63 y (29–77)	58/42	*BRAF* *NRAS*	ddPCR	0.05%	88	response, PFS, OS	Positive value: >1% *BRAF/NRAS*mut-positive droplets
Diefenbach 2020 [95]	74	III IV	NR	treatment naïve	61 y (23–88)	74/26	30-genes melanoma custom panel *BRAF, NRAS,-KIT*	NGS ddPCR	0.2% (NGS)	84	stage with cfDNA input	Positive value: >1% *BRAF/NRAS/KIT* mutation-positive droplets
Khagi 2017 [103]	69 (10 melanoma)	IV	NR	anti-PD-1	56 y (21–85)	62.3/37.7	73-genes panel	NGS	0.1%	91	PFS, OS, response (SD ≥ 6, PR, CR)	Prognostic stratification: VUS > 3 alterations better outcome
Forschner 2019 [105]	35	IV	213 d	anti-PD-1 +/− anti-CTLA-4	55 y (17–79)	54/46	*BRAF* 710-tumor associated genes	NGS ddPCR	NR	NR	response, OS	Prognostic stratification: TMB high > 23.1 better outcome
Bratman 2020 [62]	94 (10 melanoma)	IV	13.8 mo	anti-PD-1	59 y	48/62	tumor specific alterations	WES and personalized ddPCR-NGS	0.004%	98	PFS, OS, CBR	Positive value: 0.07 MTM
Eroglu 2023 [108]	29 (cohort B) 10 (cohort C)	III IV	14.2 mo (cohort B) 14.67 mo (cohort C)	cohort B: nivo +/− ipi or ICI + agent cohort C: after planned completion of ICI for MM disease	64 y (39–89) (cohort B) 66 y (51–85) (cohort C)	69/31 (cohort B) 70/30 (cohort C)	tumor specific alterations	WES and personalized ddPCR-NGS	0.004%	90 (cohort B) 10 (cohort C)	PFS	Positive value: 0.07 MTM
Schroeder 2024 [109]	87	III-IV resected *n* = 22 III-IV unresectable *n* = 65	NR	adjuvant nivo/pem *n* = 22 Systemic treatment nivo + ipi *n* = 65	64 y (56–76)	44/56	700-gene panel	targeted NGS	NR	87	LDH, S100, PET/CT MTV, PFS, OS	Positive: ≥3 tumor variants
Gangadhar 2018 [125]	25	III IV	NR	BRAF/MEKi or anti-PD-1 +/− anti-CTLA-4	57.6 y	72/28	61-gene panel	NGS	1%	48	tumor burden	NR
Olbryt 2021 [126]	22	IV	NR	BRAF/MEKi or anti-PD-1 +/− anti-CTLA-4	52 y	40/60	52-gene panel	NGS	0.1%	NR	LDH	NR

Abbreviations: pem, pembrolizumab; w, weeks.

### 2.3. ctDNA in the Cerebrospinal Fluid for Monitoring Intracranial Response

A major limitation of ctDNA analysis is its unsuitability as a biomarker for tracking evolution of tumors that metastasize to the brain. Patients with brain tumors often have low and/or undetectable ctDNA, suggesting that the blood–brain barrier may significantly impact the release of tumor DNA into the systemic circulation, called tumor shedding.

An important study published in *Nature* [130] showed that ctDNA from CNS tumors is more abundant in CSF than in plasma and is more informative about the mutational landscape. In solid extracranial tumors with CNS metastases, less cfDNA is obtained from CSF than from plasma, although the cfDNA in CSF is predominantly ctDNA. qPCR has been used to analyze CSF genomic alterations in *HER-2/neu*-positive breast cancer, *EGFR*-amplified glioblastoma and *KRAS*-mutant lung adenocarcinoma.

More recently, ctDNA has been detected in the CSF of patients with solid tumors including melanoma, lung and breast cancer using NGS and ddPCR. To date, few studies have shown that ctDNA kinetics can be derived from CSF, allowing the assessment of tumor dynamics from CNS metastases and/or leptomeningeal disease of melanoma patients [131,132]. In one study that included only eight patients with *BRAF* V600E/K-mutated melanoma, ctDNA was quantified in CSF using ddPCR. Although the study included few patients, CSF ctDNA levels reflected tumor burden and response to therapy [131]. In another study, ctDNA from CSF from only one patient with advanced melanoma with leptomeningeal dissemination, treated with whole-brain radiotherapy and BRAFi, was analyzed using ddPCR [132]. The MAF in CSF ctDNA changed according to patient clinical evolution. In one patient, MAF in CSF ctDNA was 53% at baseline and decreased to 0% at the time of clinical response corresponding to symptom relief. Afterwards, three months following clinical improvement, the patient presented a new worsening of neurological symptoms, and MAF in CSF ctDNA was again detected at high levels. Also, WES was performed to examine the mutation profiles of leptomeningeal disease, and a PTEN mutation was found before response and at disease progression.

It is important to note that the presence of ctDNA in the CSF alone is not sufficient to diagnose LMD, which requires viable, dividing tumor cells in the leptomeningeal space. Parenchymal brain lesions close to the leptomeningeal or ventricular spaces can shed ctDNA into the CSF without tumor cells invading these areas. Indeed, tumor-related mutations (e.g., *EGFR*, *BRAF*, *KRAS*, *PTEN*, *MET*) have been found in up to 63% of patients with brain metastases but no apparent LMD [130,133].

A recent study of melanoma patients with confirmed LMD showed a strong correlation between genomic alterations detected by ddPCR, the presence of circulating tumor cells in the CSF and MRI abnormalities. Notably, around 30% of CSF samples that were cytologically negative or indeterminate for tumor cells tested positive for ctDNA using ddPCR, offering valuable diagnostic insights beyond conventional methods [134]. The detection of ctDNA in CSF could therefore assist in diagnosing LMD and potentially in monitoring treatment efficacy in this context. However, due to the invasive nature of the CSF collection, this approach might be limited to select patient groups, for example, those with a cerebrospinal fluid diversion system. It is also important to note that the presence of ctDNA in CSF does not necessarily implicate LMD, as it may also indicate CNS metastases without evidence of LMD.

## 3. Conclusions and Future Directions

This review highlights promising and consistent findings on the potential applications of blood-derived ctDNA in melanoma patients. Initial studies evaluating the utility of ctDNA in advanced melanoma employed less sensitive methods, such as standard qPCR, which required mutations to be present in >1% of ctDNA to be detectable an analytical sensitivity too low to be practical in this setting [40].

In contract, the adoption of ddPCR for detecting common melanoma mutations including *BRAF*, *NRAS*, *KIT*, and *TERT* significantly improved sensitivity, allowing ctDNA to be distinguished from total cfDNA in approximately 80% of melanoma patients [72]. This advancement has enabled key applications: the identification of MRD in high-risk resected patients and the monitoring of response in advanced-stage patients undergoing standard therapies [64,72]. These include ICI, such as anti-PD-1 monotherapy, anti-PD-1 plus anti-CTLA-4, as well as targeted therapies like BRAF/MEKi combinations. However, data on the use of ctDNA for monitoring patients treated with anti-PD-1 plus anti-LAG-3 therapies remain limited. Most studies have demonstrated a clear correlation between tumor burden as assessed by imaging techniques, such as CT-scan or 18F-FDG PET/CT and ctDNA levels. Furthermore, ctDNA often provides a more reliable indication of patient outcomes across different therapeutic strategies compared to radiological assessments.

Two randomized trials in resected high-risk patients provided a ctDNA detection rate between 11% and 16% using a different technique, a ddPCR and NGS customized test, Signatera^TM^, with which an even higher detection rate was observed, at up to 40%, in more advanced stages, such as stage IIID [82,83,84]. However, although ddPCR might be more cost-effective in this context, there must be one driver mutation previously identified, and this does not occur in >20% or more melanoma patients. This aspect, and the low shedding rate in the localized setting, make it more challenging to rely on ctDNA results for monitoring MRD if we do not use a high-sensitivity technique that analyses a broader mutation repertoire, which is the case for bespoke NGS assays.

As previously disclosed, the quantification of *BRAF* V600E ctDNA levels at baseline and during treatment with BRAF/MEKi can be used to predict treatment outcomes [104,110]. In particular, two studies [104,114] found that patients with decreased levels of ctDNA had better PFS than those with maintained ctDNA levels after starting treatment with BRAF+/−MEKi, even if they were defined as non-responders according to radiological assessment. Also, in these two studies, low baseline ctDNA concentrations were associated with better response to therapy and longer PFS. Another important study, including 732 MM patients treated with BRAF+/−MEKi inside four clinical trials [110], is in agreement with previous findings that the presence of mutant *BRAF* ctDNA at baseline is associated with worse clinical outcomes in MM patients treated with BRAF/MEKi. The study also found that patients with detectable ctDNA levels after 4 weeks of treatment had worse PFS and OS compared to patients with undetectable ctDNA.

Alongside this, the detection of ctDNA was also proved to be suitable for monitoring patients receiving ICI. In a study [47] that included MM patients treated with ICI, baseline ctDNA concentration reflected tumor burden, and increased ctDNA concentration correlated with shorter OS and risk of death. On the other hand, low levels of ctDNA in treatment-naïve patients were associated with better response and improved PFS in patients treated with ipilimumab, nivolumab, and pembrolizumab.

Another application of ctDNA in melanoma patients is the discrimination of true progression from pseudoprogression. An important study evaluating this issue [71] reported a sensitivity of 90% and specificity of 100%, showing higher positive and negative predictive values than LDH. Other applications explored are the detection of molecular residual disease in advanced patients who have a complete response and discontinue treatment, as well as the detection of some resistance mechanisms such as the appearance of the *NRAS* mutation during the treatment with BRAF/MEKi or the amplification of the *BRAF* gene itself. Importantly, comparing to other biomarkers, ctDNA was also found to be a better prognostic factor than LDH across several studies [47,116].

On the other hand, NGS approaches have subsequently been applied as a ctDNA method. They have the advantage, as previously said, of analyzing many genes at the same time, and obtain a global vision of the tumor mutational scenario of each patient. The studies presented a concordance of tumor tissue with ctDNA NGS of 80%, but low analytical sensitivities in standard NGS [48,49]. However, this changes with the modified NGS that currently exist, which provide a similar sensitivity to ddPCR and allow testing a broader range of patients regardless of their tumor molecular landscape. The more sensitive NGS techniques available today can detect ctDNA in advanced melanoma patients in up to 90% of patients. Standard NGS techniques represent a significant challenge for any library preparation methodology required for sequencing, and its use requires specific strategies such as barcoding and target capture to reach an analytical sensitivity of <1%. However, new techniques with high sensitivity and specificity are currently being tested, such as the Signatera^TM^ test, which initially performs WES comparing peripheral blood and tumor tissue, identifying the 16 most frequent mutations in the tumor and then monitoring them in blood using ddPCR [97]. This allows performing ctDNA analysis in patients with MM without driver mutations, apart from having a more real knowledge of tumor molecular scenario from each patient at distinct points in the clinical course of disease.

Noteworthy, NGS methods have enabled the analysis of multiple genes simultaneously and have contributed significantly to the identification of resistance mechanisms to BRAF/MEKi in clinical studies. For example, studies have identified the presence of genomic alterations such as *NRAS* and *PIK3CA* mutations, *BRAF* amplification, and activation of the MAPK pathway as driving acquired resistance mechanisms in patients progressing on BRAF/MEKi [52,114,115]. The identification of these resistance mechanisms can help guide the selection of subsequent treatment strategies for these patients. The detection of *NRAS* Q61K/R mutations has been reported in MM patients who progressed on BRAF/MEKi treatment, and this was found to be positively correlated with an increase in *BRAF* mutant ctDNA levels. To this purpose, ddPCR analysis of ctDNA has been shown to be more sensitive than NGS ctDNA sequencing in detecting *NRAS* mutations, but NGS analysis can provide additional information such as TMB status.

Finally, it is important to emphasize the limitations of peripheral blood ctDNA to reflect the tumor burden in patients with MM with exclusively intracranial disease or predominantly subcutaneous or cutaneous metastases. In many of the studies cited, it has been observed that ctDNA is not related to the response to treatments in these patients; a very clear example is the absence of ctDNA elevation at the time of progression if it only takes place at the CNS level [71,96]. In many of the studies cited, it has been observed that ctDNA is not well correlated to the response to treatment in those patients; a very clear example is the absence of ctDNA elevation at the time of progression if it only takes place at the CNS level. To date, studies including a limited number of patients see the determination of ctDNA in CSF as feasible, especially in patients with leptomeningeal involvement. However, it is a bloody technique to perform repeatedly if it is not necessary for the well-being of patients, and at the same time, its detection can be affected by the release of ctDNA in the CSF, often altered by the blockage of the blood–brain barrier caused at the same time by the presence of brain metastasis. However, based on the limited data available to date, although cerebrospinal fluid contains a lower concentration of cfDNA than plasma, this corresponds to a higher percentage of ctDNA [130,134]. More sensitive techniques that analyze a broader repertoire of somatic variants, such as customized NGS assays, could be an alternative method for peripheral blood ctDNA analysis in patients with primarily CNS disease. However, this needs to be explored in randomized clinical trials.

Finally, the standardization of pre-analytical procedures such as time of blood collection, type of blood collection tubes and units for each analytical method in which results should be analyzed and reported, and the establishment of specific thresholds that might be considered clinically relevant should be a future priority focus of peripheral blood ctDNA research towards clinical implementation. In conclusion, most of the studies presented include a small number of patients, but ctDNA data as a prognostic factor and predictor of response to both ICI and BRAF/MEKi are mostly consistent with modified ddPCR and NGS. We believe that the next step would be its validation in prospective clinical trials evaluating specific applications, such as molecular residual disease in patients with advanced disease who have presented a prolonged CR, where the discontinuation of treatment with both ICI and BRAF/MEKi is considered, or the adaptation of treatment in advanced settings based on ctDNA evolution, such as the currently ongoing CacTUS phase II trial, which aims to switch treatment based on ctDNA response in *BRAF*V600-mutant melanoma patients. There is also an urgent need to standardize the preanalytical process for ctDNA extraction and analytical techniques to establish relevant thresholds.

## Figures and Tables

**Figure 1 ijms-26-00861-f001:**
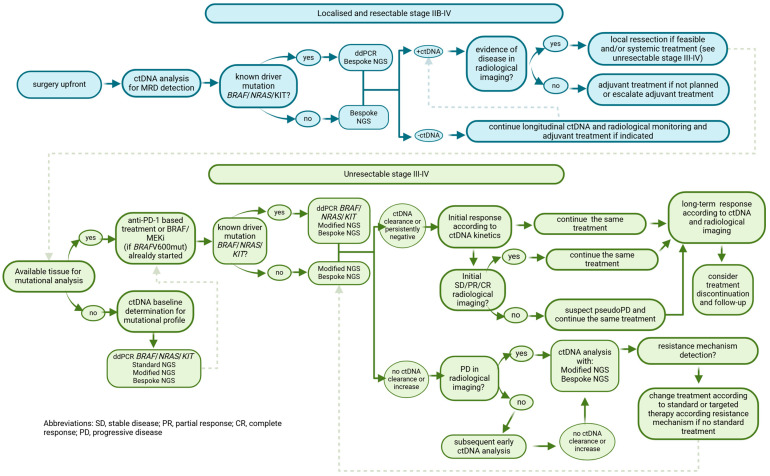
Flow chart on how to use ctDNA in clinical decision-making in melanoma patients.

**Figure 2 ijms-26-00861-f002:**
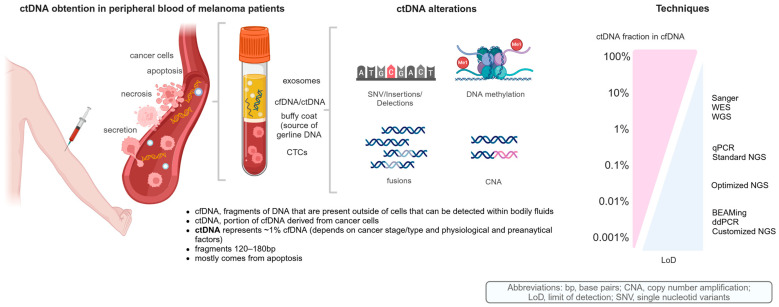
ctDNA in melanoma patients: tumor-derived components in peripheral blood, DNA-based alterations and analytical sensitivity from current techniques for ctDNA analysis.

**Figure 3 ijms-26-00861-f003:**
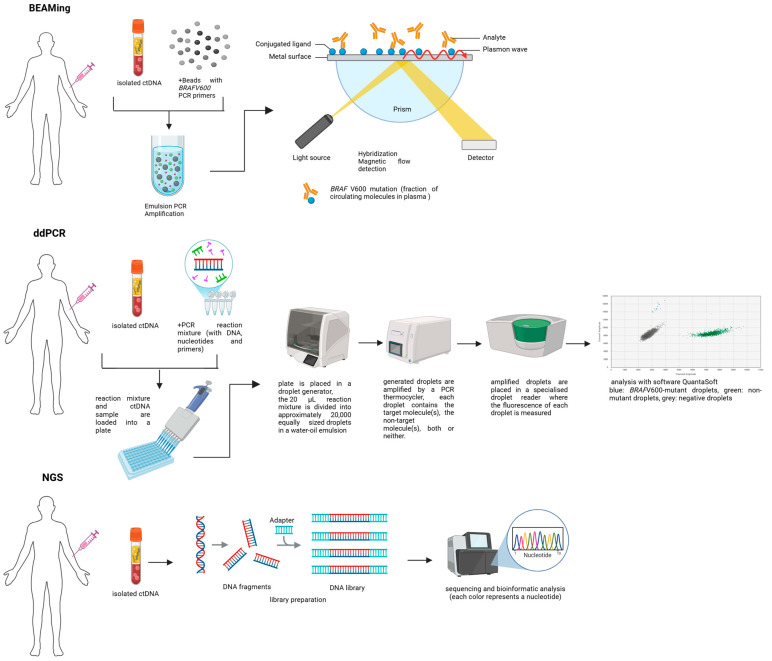
BEAMing, ddPCR and NGS schematic flowchart.

**Table 2 ijms-26-00861-t002:** Analytical techniques for ctDNA.

	Strengths	Weaknesses	LoD (Assay Sensitivity)
Standard PCR-based techniques	Selective amplification of known DNA sequences Cost-efficient and rapid	Particular sequences flanking the sequence of interest must be known, and the process is limited to a single mutation per test Danger of contamination Amplification errors will be further amplified	0.1% qPCR *BRAF* 0.005% allele-specific qPCR *BRAF*
ddPCR	Cost-efficient and rapid High sensitivity, accuracy and reproducibility Quantitative: mutant and wild-type copy number	Particular sequences flanking the sequence of interest must be known, and the process is limited to 1–2 mutations per test Danger of contamination Amplification errors will be further amplified	0.005% *BRAF*
BEAMing	High sensitivity, accuracy and reproducibility	Particular sequences flanking the sequence of interest must be known, and the process is limited to a single mutation per test Danger of contamination Amplification errors will be further amplified	0.01% *BRAF*
Standard NGS	Several genomic alterations in parallel allow tumor mutational burden analysis Greater mutational landscape information	Semiquantitive: variant allele frequency Higher cost, bioinformatic turn-out time Low sensitivity	1% targeted NGS 0.1% NGS with molecular barcode
Modified NGS Amplicon deep sequencing Hybrid-capture deep sequencing	Higher sensitivity than standard NGS Several genomic alterations in parallel allow tumor mutational burden analysis Greater mutational landscape information Detection of sub-clonal mutations or changes in clonal composition over time	Semiquantitive: variant allele frequency Higher cost, bioinformatic turn-out time	0.01% modified NGS
Bespoke assays (WES)/(WGS) + ddPCR)	High sensitivity and specificity Quantitative: mean tumor molecules (MTM)/mL Overcomes non-tumoral cfDNA contamination CHIP Several genomic alterations in parallel allow tumor mutational burden analysis Greater mutational landscape information Detection of sub-clonal mutations or changes in clonal composition over time	Higher cost, bioinformatic turn-out time Requires large amount of tumor tissue	0.004% Signatera^TM^

Abbreviations: CHIP, clonal hematopoiesis of indeterminate potential; MTM/mL, mean tumor molecules; WES, whole exome sequence; WGS, whole genome sequence.

## Data Availability

Not applicable.

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
