# Peer review of "Detection of Circulating Tumor DNA in Liquid Biopsy: Current Techniques and Potential Applications in Melanoma"

_ijms, 2025, doi:10.3390/ijms26020861_

Round 1
Reviewer 1 Report
Comments and Suggestions for Authors
The authors present a broadly detailed review of detection of ctDNA in melanoma patients at different disease stages and when used as follow-up after different type of treatment. A good summary of the relevant methods such as NGS, qPCR, BEAMING, and ddPRC is provided. The review is clearly written and the key results of many studies are summarized. Some suggestions for improvement are:
(1) earlier in the review, and perhaps in a table, present the clinical characteristics of the different stages of melanoma
(2) Discuss a bit more about the timing of ctDNA measurement after sample gathering in light of the mentioned variable instability of ctDNA in the bloodstream.
(3) Present a table that summarizes for the different detection methods their sensitivity and strengths and weaknesses and required sample sizes and reliability.
Author Response
Reviewer 1:
The authors present a broadly detailed review of detection of ctDNA in melanoma patients at different disease stages and when used as follow-up after different type of treatment. A good summary of the relevant methods such as NGS, qPCR, BEAMING, and ddPRC is provided. The review is clearly written and the key results of many studies are summarized. Some suggestions for improvement are:
(1) earlier in the review, and perhaps in a table, present the clinical characteristics of the different stages of melanoma
I have elaborated a table with clinical characteristics of different stages of melanoma and current standard management.
(2) Discuss a bit more about the timing of ctDNA measurement after sample gathering in light of the mentioned variable instability of ctDNA in the bloodstream.
I have discussed further the aspects about sample gathering and preanalytical conditions.
(3) Present a table that summarizes for the different detection methods their sensitivity and strengths and weaknesses and required sample sizes and reliability.
I have elaborated a table with strenghs and weaknesses of all techniques.
Reviewer 2 Report
Comments and Suggestions for Authors
Major revisions
The manuscript provides a thorough and detailed review of ctDNA in melanoma; however, the organization of sections could be improved. For instance, the transition between discussions on ddPCR and NGS appears abrupt. Consider restructuring to group technologies under broader headings, such as "Analytical Techniques" and "Clinical Applications."
Figures and tables are underutilized. Visual summaries of techniques, sensitivity comparisons, and clinical applications (e.g., a flowchart on how to use ctDNA in clinical decision-making) would enhance readability.
While the manuscript covers technical methods comprehensively, some explanations are too dense for a broad audience. For example, the sections on droplet digital PCR (ddPCR) might benefit from simplified descriptions or schematic illustrations to aid comprehension for readers not familiar with the technology.
While the manuscript discusses multiple techniques, it lacks a critical comparison of their advantages and limitations in specific clinical settings. A comparative table summarizing this information would add value.
The manuscript acknowledges the promise of NGS but underemphasizes emerging technologies like Signatera™ and Guardant360®. A dedicated section on emerging platforms, their unique contributions, and their limitations would strengthen the paper.
While the manuscript cites numerous studies, some references are dated. Consider integrating more recent data (2022–2024) to highlight advances in ctDNA utility, especially in immunotherapy contexts. I suggest to include the following references:
Cucciniello L, Gerratana L, Puglisi F. Liquid Biopsy, an Everchanging Balance between Clinical Utility and Emerging Technologies. Cancers (Basel). 2022 Sep 1;14(17):4277. doi: 10.3390/cancers14174277. PMID: 36077819; PMCID: PMC9454764.
Genta S, Araujo DV, Hueniken K, Pipinikas C, Ventura R, Rojas P, Jones G, Butler MO, Saibil SD, Yu C, Easson A, Covelli A, Sauder MB, Fournier C, Saeed Kamil Z, Rogalla P, Arteaga DP, Vornicova O, Spiliopoulou P, Muniz TP, Siu LL, Spreafico A. Bespoke ctDNA for longitudinal detection of molecular residual disease in high-risk melanoma patients. ESMO Open. 2024 Nov;9(11):103978. doi: 10.1016/j.esmoop.2024.103978. Epub 2024 Nov 16. PMID: 39549683; PMCID: PMC11615122.
Schroeder C, Gatidis S, Kelemen O, Schütz L, Bonzheim I, Muyas F, Martus P, Admard J, Armeanu-Ebinger S, Gückel B, Küstner T, Garbe C, Flatz L, Pfannenberg C, Ossowski S, Forschner A. Tumour-informed liquid biopsies to monitor advanced melanoma patients under immune checkpoint inhibition. Nat Commun. 2024 Oct 9;15(1):8750. doi: 10.1038/s41467-024-52923-0. PMID: 39384805; PMCID: PMC11464631.
Azam A, Us Saqib HW. Redefining melanoma surveillance: The controversial utility of serum S100B and the potential of liquid biopsy. Eur J Surg Oncol. 2024 Dec;50(12):108695. doi: 10.1016/j.ejso.2024.108695. Epub 2024 Sep 18. PMID: 39317100.
Minor Revisions
Some sentences are overly long and difficult to follow. For example, the sentence in the "Introduction" about ctDNA detection using PCR-based methods can be split into two for better clarity.
Several references are cited repeatedly but lack accompanying context in the discussion. For example, AVAST-M and CheckMate 915 trials are mentioned but not critically analyzed for their methodological differences.
The term "liquid biopsy" is used interchangeably with "ctDNA analysis." While related, clarify when each term is intended to describe broader versus specific applications.
The manuscript references ctDNA in units such as "copies/mL" and "percentage of total DNA." Ensure consistent metrics or provide a rationale for varying units.
Figure 1 could benefit from clearer annotations, particularly on how ctDNA is distinguished from cfDNA.
Include a section that discusses future research priorities, such as standardizing ctDNA thresholds for treatment decisions and addressing limitations in detecting brain metastases.
Author Response
Reviewer 2:
Comments and Suggestions for Authors
Major revisions
The manuscript provides a thorough and detailed review of ctDNA in melanoma; however, the organization of sections could be improved. For instance, the transition between discussions on ddPCR and NGS appears abrupt. Consider restructuring to group technologies under broader headings, such as "Analytical Techniques" and "Clinical Applications."
I have restructured this information in the review, splitting it into two sections and improving the content and readability of the text.
Figures and tables are underutilized. Visual summaries of techniques, sensitivity comparisons, and clinical applications (e.g., a flowchart on how to use ctDNA in clinical decision-making) would enhance readability.
I have prepared a table showing the strengths and weaknesses of all techniques and added a proposed flowchart for clinical applications in different contexts of stage, known mutation profile and clinical variable to assess their correlation with ctDNA. I have also added a figure summarising the flowchart of the BEAming, ddPCR and NGS techniques to improve compressibility.
While the manuscript covers technical methods comprehensively, some explanations are too dense for a broad audience. For example, the sections on droplet digital PCR (ddPCR) might benefit from simplified descriptions or schematic illustrations to aid comprehension for readers not familiar with the technology.
While the manuscript discusses multiple techniques, it lacks a critical comparison of their advantages and limitations in specific clinical settings. A comparative table summarizing this information would add value.
I have elaborated a chart with strenghs and weaknesses of all techniques and a figure with ddPCR and BEAMING and NGS simplified flowchart. The manuscript acknowledges the promise of NGS but underemphasizes emerging technologies like Signatera™ and Guardant360®. A dedicated section on emerging platforms, their unique contributions, and their limitations would strengthen the paper.
I have added an special section about these more technically advanced techniques.
While the manuscript cites numerous studies, some references are dated. Consider integrating more recent data (2022–2024) to highlight advances in ctDNA utility, especially in immunotherapy contexts. I suggest to include the following references:
Cucciniello L, Gerratana L, Puglisi F. Liquid Biopsy, an Everchanging Balance between Clinical Utility and Emerging Technologies. Cancers (Basel). 2022 Sep 1;14(17):4277. doi: 10.3390/cancers14174277. PMID: 36077819; PMCID: PMC9454764.
Genta S, Araujo DV, Hueniken K, Pipinikas C, Ventura R, Rojas P, Jones G, Butler MO, Saibil SD, Yu C, Easson A, Covelli A, Sauder MB, Fournier C, Saeed Kamil Z, Rogalla P, Arteaga DP, Vornicova O, Spiliopoulou P, Muniz TP, Siu LL, Spreafico A. Bespoke ctDNA for longitudinal detection of molecular residual disease in high-risk melanoma patients. ESMO Open. 2024 Nov;9(11):103978. doi: 10.1016/j.esmoop.2024.103978. Epub 2024 Nov 16. PMID: 39549683; PMCID: PMC11615122.
Schroeder C, Gatidis S, Kelemen O, Schütz L, Bonzheim I, Muyas F, Martus P, Admard J, Armeanu-Ebinger S, Gückel B, Küstner T, Garbe C, Flatz L, Pfannenberg C, Ossowski S, Forschner A. Tumour-informed liquid biopsies to monitor advanced melanoma patients under immune checkpoint inhibition. Nat Commun. 2024 Oct 9;15(1):8750. doi: 10.1038/s41467-024-52923-0. PMID: 39384805; PMCID: PMC11464631.
Azam A, Us Saqib HW. Redefining melanoma surveillance: The controversial utility of serum S100B and the potential of liquid biopsy. Eur J Surg Oncol. 2024 Dec;50(12):108695. doi: 10.1016/j.ejso.2024.108695. Epub 2024 Sep 18. PMID: 39317100.
I have reviewed all these papers and incorporated the information I have considered that adds value to the review.
Minor Revisions
Some sentences are overly long and difficult to follow. For example, the sentence in the "Introduction" about ctDNA detection using PCR-based methods can be split into two for better clarity.
I have divided the paragraph into two sections as indicated, techniques and clinical applications.
Several references are cited repeatedly but lack accompanying context in the discussion. For example, AVAST-M and CheckMate 915 trials are mentioned but not critically analyzed for their methodological differences.
I have made an effort to discuss deeper the methodological differences of these two trials and other studies cited.
The term "liquid biopsy" is used interchangeably with "ctDNA analysis." While related, clarify when each term is intended to describe broader versus specific applications.
I have tried to improve this aspect and to be more coherent in the whole of the review.
The manuscript references ctDNA in units such as "copies/mL" and "percentage of total DNA." Ensure consistent metrics or provide a rationale for varying units.
I have tried to clarify the different metrics in which ctDNA is reported.
Figure 1 could benefit from clearer annotations, particularly on how ctDNA is distinguished from cfDNA.
I have clarified the terms cfDNA and ctDNA.
Include a section that discusses future research priorities, such as standardizing ctDNA thresholds for treatment decisions and addressing limitations in detecting brain metastases.
I have discussed these two aspects more fully in the Conclusions and Future Directions section.
Round 2
Reviewer 2 Report
Comments and Suggestions for Authors
The authors have addressed the main comments.